# Attenuated CSF-1R signalling drives cerebrovascular pathology

Conor Delaney[1] (ID), Michael Farrell[2], Colin P Doherty[3,4,5], Kiva Brennan[6], Eoin O'Keeffe[1], Chris Greene[1], Kieva Byrne[1], Eoin Kelly[3], Niamh Birmingham[7], Paula Hickey[8], Simon Cronin[7], Savvas N Savvides[9,10], Sarah L Doyle[6] (ID) & Matthew Campbell[1,5,*] (ID)

## Abstract

Cerebrovascular pathologies occur in up to 80% of cases of Alzheimer's disease; however, the underlying mechanisms that lead to perivascular pathology and accompanying blood–brain barrier (BBB) disruption are still not fully understood. We have identified previously unreported mutations in colony stimulating factor-1 receptor (*CSF-1R*) in an ultra-rare autosomal dominant condition termed adult-onset leucoencephalopathy with axonal spheroids and pigmented glia (ALSP). Cerebrovascular pathologies such as cerebral amyloid angiopathy (CAA) and perivascular p-Tau were some of the primary neuropathological features of this condition. We have identified two families with different dominant acting alleles with variants located in the kinase region of the *CSF-1R* gene, which confer a lack of kinase activity and signalling. The protein product of this gene acts as the receptor for 2 cognate ligands, namely colony stimulating factor-1 (CSF-1) and interleukin-34 (IL-34). Here, we show that depletion in CSF-1R signalling induces BBB disruption and decreases the phagocytic capacity of peripheral macrophages but not microglia. CSF-1R signalling appears to be critical for macrophage and microglial activation, and macrophage localisation to amyloid appears reduced following the induction of *Csf-1r* heterozygosity in macrophages. Finally, we show that endothelial/microglial crosstalk and concomitant attenuation of CSF-1R signalling causes remodelling of BBB-associated tight junctions and suggest that regulating BBB integrity and systemic macrophage recruitment to the brain may be therapeutically relevant in ALSP and other Alzheimer's-like dementias.

**Keywords** adult-onset leucoencephalopathy with axonal spheroids and pigmented glia (ALSP); blood; brain barrier; CSF-1; CSF-1R; IL-34
**Subject Categories** Genetics, Gene Therapy & Genetic Disease; Neuroscience

## Introduction

Alzheimer's disease (AD) is a progressive neurodegenerative disorder characterised by a gradual decline in cognitive function and is the most common cause of dementia in the elderly. Currently, there are approximately 36 million people worldwide who suffer from AD and other dementias and there are no approved medicines to prevent disease progression. The central neuropathological hallmarks of AD are as follows: (i) the extracellular accumulation of amyloid-β (Aβ) in parenchymal plaques and (ii) the formation of intracellular neurofibrillary tangles (NFTs) composed of hyperphosphorylated forms of the microtubule-associated protein tau. Additionally, up to 80% of AD patients display cerebrovascular pathologies such as cerebral amyloid angiopathy (CAA) as well as in over 30% of cases of non-Alzheimer's dementia (Matthews *et al*, 2009). CAA is characterised by an accumulation of Aβ in the parenchymal space, within vessel walls of arteries (namely the cerebral, leptomeningeal and parenchymal arteries) or around small- to medium-sized blood vessels. The underlying mechanism that leads to Aβ accumulation in this perivascular pattern is still not fully understood. It is accepted, however, that CAA can contribute to dementia onset and cognitive decline through increasing susceptibility for microbleeds, cerebral ischaemia and chronic activation of pro-inflammatory mechanisms in the surrounding parenchyma (Kinnecom *et al*, 2007; Vukic *et al*, 2009).

Recently, we have examined post-mortem brain tissues from two families with an ultra-rare condition termed adult-onset

1 Smurfit Institute of Genetics, Trinity College Dublin, Dublin 2, Ireland
2 Department of Neuropathology, Beaumont Hospital, Dublin 9, Ireland
3 Department of Neurology, Health Care Centre, St James's Hospital, Dublin 8, Ireland
4 Academic Unit of Neurology, Biomedical Sciences Institute, Trinity College Dublin, Dublin 2, Ireland
5 FutureNeuro SFI Research Centre, Royal College of Surgeons in Ireland, Dublin, Ireland
6 Trinity College Institute of Neuroscience, Trinity College Dublin 2, Dublin 2, Ireland
7 Department of Medicine, University College Cork, Cork, Ireland
8 Sligo Regional Hospital, Sligo, Ireland
9 Unit for Structural Biology, Department of Biochemistry and Microbiology, Ghent University, Ghent, Belgium
10 VIB-UGent Center for Inflammation Research, Ghent, Belgium
*Corresponding author. Tel: +353 1 8961482; Fax: +353 1 8963848; Email: matthew.campbell@tcd.ie

leucoencephalopathy with axonal spheroids and pigmented glia (ALSP). Individuals in this family presented clinically with early-onset dementia (in their forties) that had been reported within the family previously. Indeed, many of the reported symptoms for ALSP are akin to those reported in AD, namely depression, cognitive impairment, gait disturbances, speech problems and overt neurodegeneration. Adult-onset leucoencephalopathies accompanied by spheroids and pigmented glia were initially categorised as two main conditions, hereditary diffuse leucoencephalopathy with axonal spheroids (HDLS) and familial pigmentary orthochromatic leucodystrophy (POLD). While these clinical conditions have been known for nearly 40 years as distinct entities(Van Bogaert, 1936; Axelsson et al, 1984), the genetic cause of the two were reported within the past 10 years as being associated with dominant acting mutations in the colony stimulating factor-1 receptor (CSF-1R) gene (Rademakers et al, 2011). Due to the shared genetic aetiology of the conditions (Nicholson et al, 2013), they are now considered to be part of the spectrum of the same disease, which has been termed ALSP (Adams et al, 2018).

CSF-1R is activated via common structural principles by two distinct cytokine ligands, namely colony stimulating factor-1 (CSF-1) and interleukin-34 (IL-34) (Elegheert et al, 2011; Ma et al, 2012; Felix et al, 2013, 2015) which are both essential for microglial viability, development and proliferation (Askew et al, 2017; Bohlen et al, 2017; Wu et al, 2018). CSF-1R is also critical for myeloid lineage cell differentiation, as well as that of peripheral monocytes into circulating and tissue-resident macrophages (Wang et al, 2012; Rojo et al, 2019), and was recently found to be mutated in gain-of-function pathologic variants in histiocytic neoplasms (Durham et al, 2019). While CSF-1R is expressed in some cells of the adult brain under homeostasis, namely microglia, it and its ligands have been shown to be upregulated and neuroprotective in mouse disease models of AD and epilepsy (Luo et al, 2013; Schwarzer et al, 2019). In addition to CSF-1R signalling appearing to be a critical pathway for responding to CNS insult, IL-34 is secreted by neuronal cells under normal conditions and can enhance neuroprotective effects of microglia in response to stimuli such as oligomeric amyloid (Schwarzer et al, 2019). While CSF-1R has been detected previously in mouse endothelial cells within the brain and spinal cord (Jin et al, 2014), and exhibits endothelial expression in human cortical tissue (Uhlén et al, 2015), it has still not been fully confirmed that brain endothelial cells actually express the protein. In this regard, the expression pattern of CSF1R in the brain endothelium has yet to be fully characterised.

As CSF-1R is essential for the survival of the vast majority of microglial populations, it has recently become a commonly targeted receptor in drug development programmes for epilepsy and AD. CSF-1R inhibition in pre-clinical models using small molecules such as PLX3397 and PLX5622 has been shown to be effective in depleting up to 99% of microglia within 3 weeks of administration in chow (Luo et al, 2013; Rice et al, 2015; Dagher et al, 2015; Renee et al, 2015; Spangenberg et al, 2016, 2019; Bohlen et al, 2017; Singh et al, 2017; Elmore et al, 2018; Srivastava et al, 2018). Many of these studies report beneficial outcomes in mouse models of AD including reduced plaque burden (Rice et al, 2015), neuroinflammation (Spangenberg et al, 2016) and improved cognition (Dagher et al, 2015). However, these microglia-ablation studies have also produced conflicting data related to whether microglial ablation affects the rate of

Aβ deposition and cognition improvement. Furthermore, cognitive improvement has not been sustained following repopulation of the brain by microglia in these models. Whether the beneficial effects of CSF-1R inhibition are due to ameliorating a key process in the development of AD in these mice, or transiently inhibiting the overall effects of microglial inflammatory processes on cognition, has not been established.

Here, we have observed Aβ-associated cerebrovascular pathology accompanied by BBB breakdown in 2 separate families with previously unreported and different dominant acting mutations in the kinase region of CSF-1R. These mutations lead to a profound loss of function of CSF-1R and attenuated signalling, rendering affected individuals essentially haploinsufficient for CSF-1R. Surprisingly however, we found that microglial cells heterozygous for Csf-1r retain robust phagocytic ability, but macrophages with depleted CSF-1R signalling are significantly compromised. We show that dysregulated CSF-1R-dependent endothelial/microglial cell crosstalk induces re-modelling of the BBB in mouse and human and suggests that regulating BBB integrity while promoting macrophage recruitment to the brain may be therapeutically viable in ALSP and other AD-like dementia. Added to this, autologous bone marrow transplants could show real and meaningful clinical readouts in ALSP patients.

# Results

## Dominant mutations in CSF-1R induce cerebral amyloid angiopathy (CAA) and cerebrovascular pathology

We identified 2 separate families showing clinical evidence of the ultra-rare condition ALSP. Concomitantly, we confirmed the presence of mutations within the kinase region of the CSF-1R gene, ΔA781_N783 in one family (Fig 1A) and a P824R mutation in another (Fig 1B). Post-mortem analysis of the brains of 3 genetically diagnosed ALSP cases revealed extensive cerebral atrophy (Figs 1C and D and EV1). Widespread white matter Wallerian degeneration was also observed (Fig 1E). Very extensive axonal spheroids were apparent when sections of brain tissue were stained for neurofilament (Fig 1F). Furthermore, widespread cerebral amyloid angiopathy (CAA), involving meningeal and intraparenchymal blood vessels, was also evident in tandem with severe arteriosclerosis in subcortical white matter (Fig 1G and H). An identical pathology was observed in all 3 post-mortem cases of ALSP. Perivascular accumulation of phosphorylated Tau (P-tau) (Fig 1I and J) and amyloid-beta (Aβ) was clearly evident, and observation of such unique pathologies in a single disease points towards a deficit in the underlying processes through which these proteins aggregate and/or are cleared across the endothelium that constitutes the BBB.

## Mutations in CSF-1R attenuate kinase activity and signalling

ALSP is primarily inherited in an autosomal dominant manner, and in these particular families, it results from either a 9-base-pair in-frame deletion in the CSF-1R gene, (c.2342_2350del) leading to an Ala781-Arg782-Asn783 deletion, or a single point mutation (c.2741C > G) leading to a Pro824Arg substitution. These are

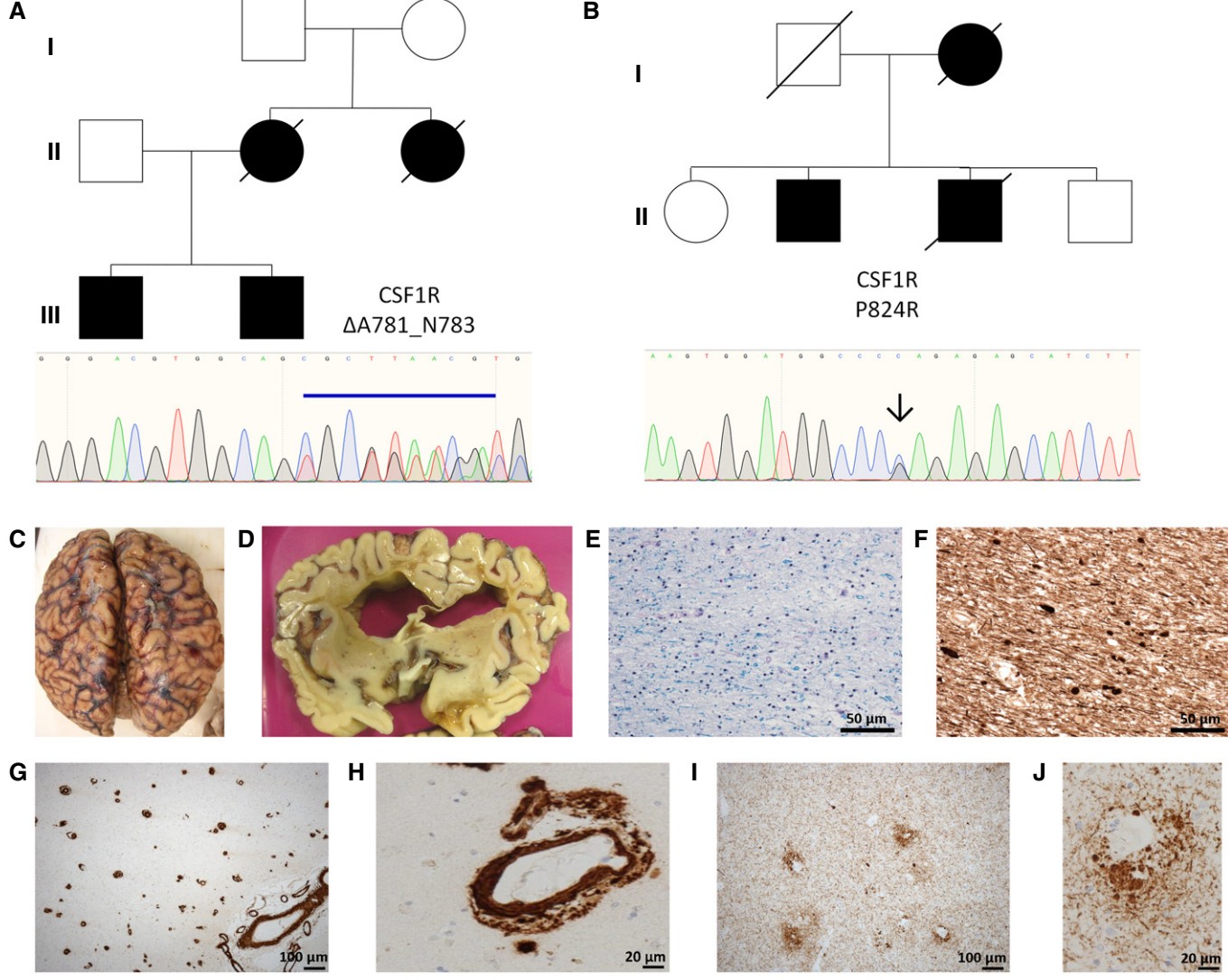

**Figure 1. Identification of familial variants in *CSF-1R*.**

A, B  Family pedigrees for *CSF-1R* variants ΔA781_N783 and P824R, with Sanger sequencing confirmations for each variant below.
C  Cystic encephalomalacia evident in the right inferior frontal lobe secondary to a historical intraparenchymal haemorrhage.
D  Extensive atrophy evident in frontal lobe.
E  Secondary de-myelination observed with Luxol-Fast Blue stain.
F  Wallerian degeneration observed with spheroids apparent following phosphorylated neurofilament staining.
G, H  Profound cerebral amyloid angiopathy (CAA), involving meningeal and intraparenchymal blood vessels.
I, J  Extensive phospho-tau staining displaying a perivascular pattern of deposition.

previously unreported mutations in *CSF-1R* and based on sequence considerations both mutations would be expected to be found in the intracellular kinase domain of CSF-1R. Analysis of the structural context of these mutations using the crystal structure of the kinase domain of human CSF-1R (pdb entry 2OGV) (Walter *et al*, 2007) reveals that the mutated sites localise within highly conserved sequence cassettes in the C-lobe of the kinase region of CSF-1R (Fig 2A). In the case of the Ala781-Asn783 deletion mutation, the 3 amino acids are critical for comprising a unique left-handed helical secondary structural element within the active site of CSF-1R (Fig 2 A). This allows projection of Arg782 to interact with the catalytic

residue Asp778 and the activation loop of CSF-1R. Furthermore, Ala781 makes key interactions with a hydrophobic core of the C-lobe (Fig 2A). Therefore, the Ala781-Asn783 deletion mutation would be predicted to be a loss-of-function mutation and to exhibit compromised protein stability. Similarly, Pro824 is found at the end of the activation loop of CSF-1R and nestles into a strictly hydrophobic pocket to provide structural anchoring and stability. Thus, a Pro824Arg mutation would be expected to be overwhelmingly incompatible with such an environment leading to destabilisation of both the activation loop of CSF-1R and part of the hydrophobic core of the C-lobe of the CSF-1R kinase domain (Fig 2A).

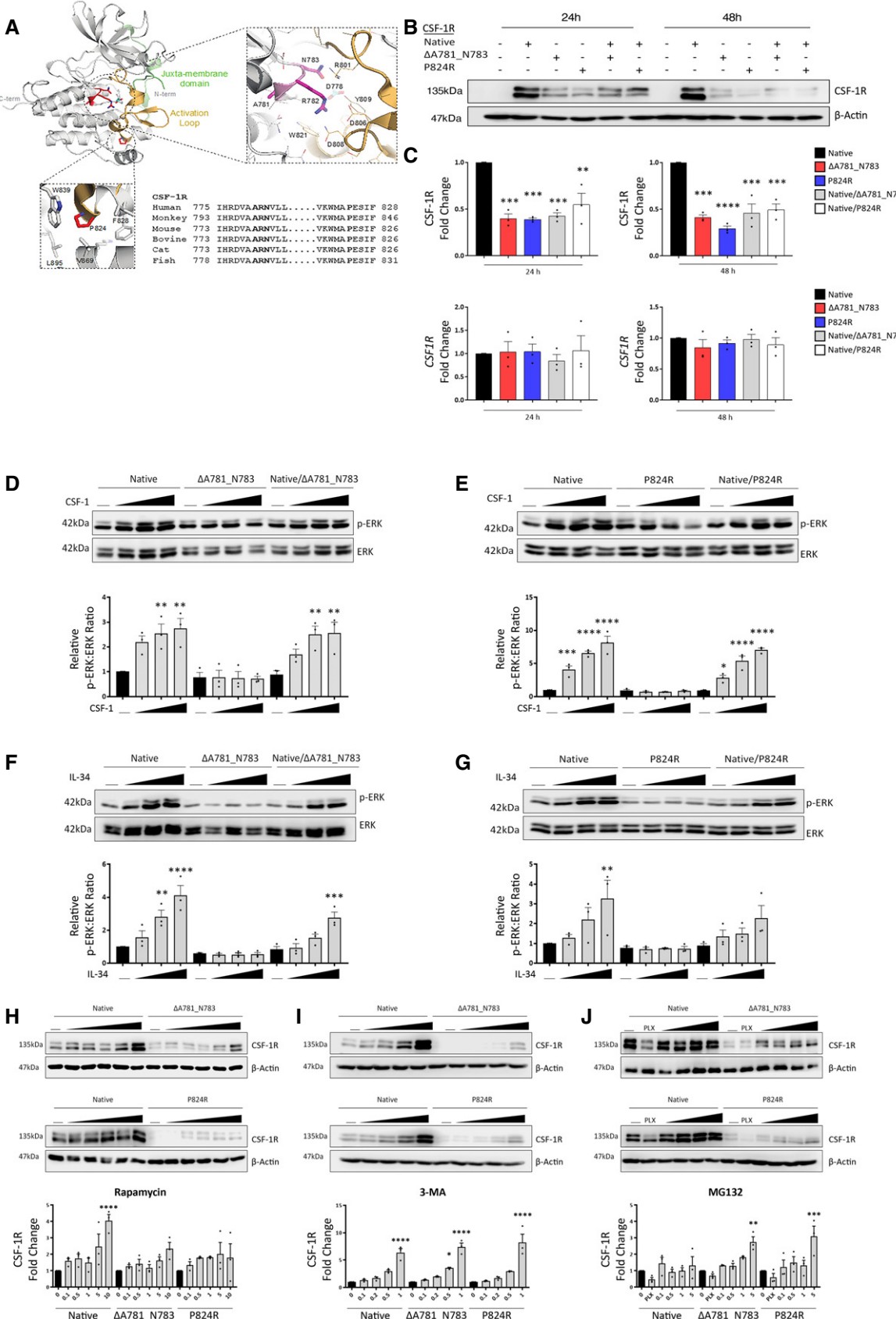

**Figure 2.**

**Figure 2.  The ΔA781_N783 and P824R variants in CSF-1R are loss of function and have conserved cellular processing.**

A       Structure-sequence analysis and context of the CSF-1R ΔA781_N783 and P824R variants. Sequences used in the alignment correspond to Uniprot sequences for CSF-1R as follows: Human P07333, Cynomolgus Monkey A0A2K5WG91, Mouse P09581, Bovine A7Z067, Cat P13369, Zebrafish Q9I8N6.

B       Western blot for CSF-1R and Actin in HEK293 cells transfected with native and variant CSF-1R as indicated.

C       Densitometry (top) and qPCR analysis of native and variant CSF-1R expression in HEK293 cells relative to native-transfected at 24 and 48 h (****$P < 0.0001$, ***$P < 0.0005$, , **$P < 0.005$ one-way ANOVA with Sidak's post-test for multiple comparison, $n = 3$ biological replicates, error bars indicate SEM).

D–G   Western blot for phosphorylated and total ERK in HEK293 cells transfected with native and ΔA781_N783 (left) or P824R (right) CSF-1R and treated with CSF-1 (D, E) or IL-34 (F, G) for 10 min. Horizontal line indicates untreated cells, with increasing concentrations 10, 50 and 100 ng/ml. Corresponding densitometry displays the ratio of phospho-ERK to total ERK, normalised to the ratio of the untreated control for the native receptor (****$P < 0.0001$, ***$P < 0.0005$, **$P < 0.01$, *$P < 0.05$, one-way ANOVA with Sidak's post-test for multiple comparison, $n = 3$ biological replicates, , error bars indicate SEM).

H       Western blot for CSF-1R in HEK293 cells transfected with native and ΔA781_N783 (top) or P824R (bottom) CSF-1R and treated with rapamycin. Horizontal line indicates untreated cells, with concentrations 0.1, 0.5, 1, 5, 10 μM. Corresponding densitometry (below) displays the fold change of CSF-1R relative to the untreated control for each transfection (****$P < 0.0001$, one-way ANOVA with Sidak's post-test for multiple comparison to untreated, $n = 3$ biological replicates, error bars indicate SEM).

I        Western blot for CSF-1R in HEK293 cells transfected with native and ΔA781_N783 (top) or P824R (bottom) CSF-1R and treated with 3-methyladenosine. Horizontal line indicates untreated cells, with increasing concentrations 0.1, 0.2, 0.5, 1 mM. Corresponding densitometry (below) displays the fold change of CSF-1R relative to the untreated control for each transfection (****$P < 0.0001$, *$P = 0.0114$, one-way ANOVA with Sidak's post-test for multiple comparison to untreated, $n = 3$ biological replicates, error bars indicate SEM).

J        Western blot for CSF-1R and Actin in HEK293 cells transfected with native and ΔA781_N783 (top) or P824R (bottom) CSF-1R and treated with MG132 or PLX3397. Horizontal line indicates untreated cells, 20 μM PLX3397, with increasing concentrations 0.1, 0.5, 0.1, 5 μM. Corresponding densitometry (below) displays the fold change of CSF-1R relative to the untreated control for each transfection (***$P = 0.0002$, **$P = 0.002$, one-way ANOVA with Sidak's post-test for multiple comparison to untreated, $n = 3$ biological replicates, error bars indicate SEM).

Generation of a native CSF-1R gene in tandem with genes containing the 2 mutations clearly showed that protein expression levels of CSF-1R were decreased in both mutations, with no accompanying decrease in transcriptional activity (Fig 2B and C). Localisation of CSF-1R is still conserved to the plasma membrane even in the presence of a CSF-1R mutation, suggestive of no dominant-negative effect of the mutation (Fig EV2). The concept of mutant CSF-1R aggregating and accumulating in cells was then explored using a cell stress array approach. In cells over-expressing CSF-1R with the P824R variant, out of 84 cell stress-related genes examined only *DDIT3* and *CXCL8* had significantly altered gene expression values when compared to cells expressing native CSF-1R (Appendix Fig S1).

Importantly however, while stimulation of cells expressing native CSF-1R with recombinant CSF-1 (Fig 2D and E) or IL-34 (Fig 2F and G) at increasing doses caused increased and dose-dependent ERK phosphorylation, no ERK phosphorylation was evident in cells expressing mutant CSF-1R. p-ERK was observed in cells co-expressing native and mutant CSF-1R, suggestive of retained function of native CSF-1R in the presence of variant CSF-1R. The IL-34-mediated increase in p-ERK was reduced in the heterozygous context; however, in response to CSF-1, similar levels of ERK phosphorylation were observed in native-only and native-variant transfected cells. In order to rule out the possibility that absence of CSF-1R signalling is due to the reduced level of variant CSF-1R, we pretreated cells with MG132 which can upregulate variant CSF-1R. No signalling was observed post-stimulation, confirming loss of function of CSF-1R signalling in both mutations (Fig EV3).

In order to ascertain a putative mechanism underlying the control of CSF-1R production and turnover, we examined the expression of native or mutant CSF-1R in the presence of increasing doses of an autophagy inducer (rapamycin) (Fig 2H), an autophagy inhibitor (3-methyladenosine) (Fig 2I) or a proteasomal inhibitor (MG132) (Fig 2J). Intriguingly, during autophagy inhibition, CSF-1R protein levels are increased independent of genotype. While autophagy induction also showed a dose-dependent increase in CSF-1R levels, inhibition of the proteasome appeared to specifically regulate levels of mutated CSF-1R and not native protein (Fig 2J).

## Mutant and non-functioning CSF-1R protein decreases the phagocytic capacity of peripheral macrophages but not microglia

While CSF-1R is expressed in endothelial cells, the predominant cell types expressing this protein are immune cells of the myeloid lineage and microglia. Post-mortem analysis of ALSP patient brains showed a perivascular accumulation of CD68- and CD163-positive cells suggestive of peripheral macrophage involvement. To this end and in order to ascertain the effect of mutated CSF-1R on immune cell differentiation, we isolated PBMCs from an individual diagnosed with the ΔA781_N783 CSF-1R mutation and compared them to a normal healthy donor. Interestingly, there was a very clear depletion of T cells, CD14$^+$ macrophages in addition to B cells. Added to this, PBMCs from the ALSP patient also showed up to 90% lineage-negative cells (CD3/14/16/19/20/56) compared to a non-diseased sample (Fig 3A and B). It was also very clear that macrophages isolated from patients expressing either CSF-1R mutation displayed highly dysmorphic cells (Fig 3C). These cells accounted for 30% of all differentiated macrophages and had a significantly decreased phagocytic ability compared to macrophages isolated from healthy donors (Fig 3D–F). As CSF-1R is predominantly expressed in microglial cells in the adult mouse and human brain, we sought to determine whether the phagocytic ability of microglial cells was compromised in mice expressing only a single *Csf-1r* allele in microglia.

Unexpectedly, there was no difference between the phagocytic capacity of microglia with reduced CSF-1R signalling. Similarly, microglia which underwent pharmacological *Csf-1r* downregulation using siRNA had no change in the rate of phagocytosis (Fig 3G and H), and this was further confirmed through genetic *Csf-1r* reduction in mice. Primary microglia isolated from WT or microglial/macrophage-specific *Csf-1r*$^{+/-}$ mice displayed no altered phagocytic capacity (Fig 3J–I), together suggesting that *Csf-1r* heterozygosity was dispensable for microglial phagocytosis. However, when we examined bone-marrow-derived macrophages (BMDM's) from the same animals, the phagocytic ability of *Csf-1r*$^{+/-}$ BMDMs was significantly compromised similar to the human PBMCs, indicating a deficit in peripheral macrophage phagocytosis as opposed to

microglial cells (Fig 3L and M). These data suggest that in the human condition with *Csf-1r* haploinsufficiency, activated microglia are still robustly phagocytically active, while it is peripheral macrophages that are compromised.

### CSF-1R loss affects macrophage/microglial activation and brain recruitment

In order to ascertain whether endothelial or macrophage CSF-1R is central to immune cell response to a stimulus such as amyloid aggregation and clearance across the BBB, we injected amyloid-beta 1-42 unilaterally into the hippocampus of mice lacking a single *Csf-1r* allele in microglial/macrophage cells (Fig 4A) or endothelial cells (Fig 4C). Reduced immune cell infiltration and activation (F4/80 positivity) was only evident in the context of macrophage (Fig 4B) *Csf-1r* loss as opposed to endothelial cells (Fig 4D). This is of particular note in the context of CAA as the patrolling subset of monocytes in mice known to associate with the cerebral vasculature (Ly6C$^{lo}$, Cx3CR1$^{high}$, CCR2$^{-}$) develop in a CSF-1R-dependent manner (Yona *et al*, 2013). Targeted depletion of these

vessel-associating macrophages has previously been shown to induce an increase in Aβ load in the brain parenchyma as well as around the CNS vasculature (Michaud *et al*, 2013). Additionally, stimulation of these cells was also able to specifically reduce CAA load, highlighting the critical role of peripheral CSF-1R-dependent macrophage activity in driving amyloid clearance at the BBB (Hawkes & McLaurin, 2009).

### Endothelial/microglial crosstalk induces tight junction re-modelling and BBB disruption in ALSP

As mutations in *CSF-1R* and a lack of CSF-1R signalling can clearly induce cerebrovascular pathology in humans, we sought to elucidate the cell type responsible for mediating this phenotype. In that regard, we treated mouse microglial cells with two concentrations of the CSF-1R inhibitor PLX3397 (Hi, Lo) to observe the indirect effects of CSF-1R-inhibited microglia conditioned media (Hi-BVCM, Lo-BVCM) on endothelial cells (Fig 5A). Strikingly, we observed that cell culture media conditioned by microglia in the absence of PLX3397 could upregulate endothelial CSF-1R within 24 h of

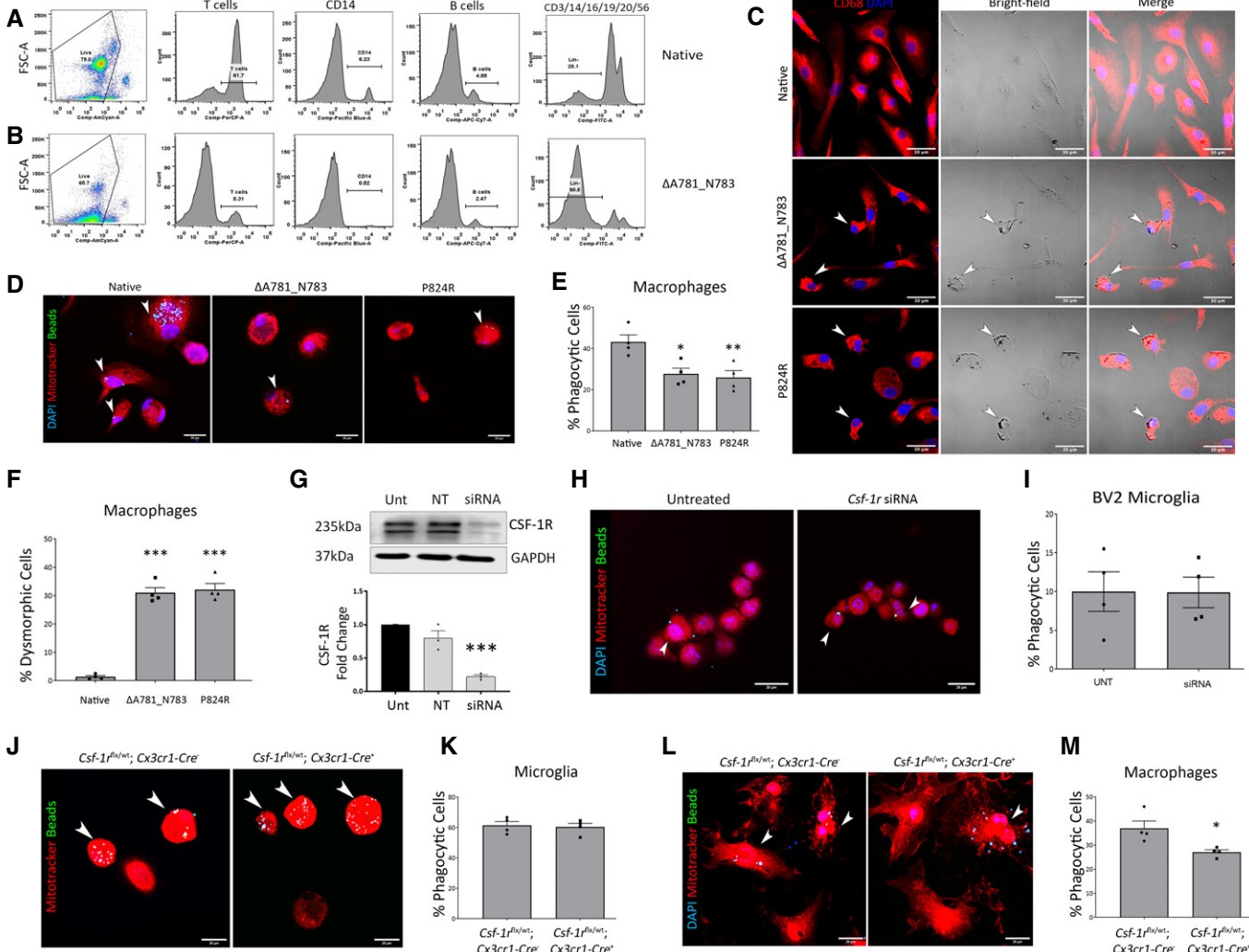

**Figure 3.**

**Figure 3.  Peripheral variant CSF-1R macrophages have altered identity and function.**

A, B     FACS of control and ΔA781_N783 PBMCs using lineage markers for CD3 (T cells), CD14 (monocytes/macrophages), CD19 (B cells), CD16, CD20 and CD56. Cells negative for all 6 makers were determined to be lineage-negative (Lin−).

C        Immunocytochemistry of macrophages differentiated *in vitro* from control, ΔA781_N783 or P824R PBMCs. Cells were stained for DAPI (blue) and CD68 (red). White arrows indicate dysmorphic macrophages.

D        Immunocytochemistry of macrophages differentiated *in vitro* from control, ΔA781_N783 or P824R PBMCs. Macrophages were stimulated with LPS and exposed to fluorescent opsonised latex beads for 1 h before fixation and quantification of bead$^+$ cells via microscopy. Cells were stained using MitoTracker (red) and DAPI (blue).

E        Quantification of phagocytic activity expressed as percentage bead$^+$ cells. $P$ values were calculated using (**$P$ = 0.008, *$P$ = 0.0141, one-way ANOVA with Tukey's multiple comparison test, $n$ = 4 assays, error bars indicate SEM).

F        Quantification of macrophages displaying an aberrant morphology as indicated in (c), displayed as percentage of dysmorphic cells per image (***$P$ < 0.0005, one-way ANOVA with Dunnett's multiple comparison test, data representative of $n$ = 2 independent differentiations, with two fields of view imaged per well, error bars indicate SEM).

G        Western blot for CSF-1R in untreated (Unt) BV2 microglia, or BV2 microglia transfected with non-targeting or *Csf-1r* targeting siRNA. Corresponding densitometry (below) representative of technical replicate of $n$ = 3 blots (one-way ANOVA with Dunnett's multiple comparison, ***$P$ = 0.0003, error bars indicate SEM).

H        Immunocytochemistry of untreated and *Csf-1r* siRNA treated BV2 microglia. BV2 microglia were stimulated with LPS and exposed to fluorescent opsonised latex beads for 1 h before fixation and quantification of bead$^+$ cells via microscopy. Cells were stained using MitoTracker (red) and DAPI (blue). Arrowheads (white) indicate bead$^+$ cells.

I        Quantification of BV2 phagocytic activity expressed as percentage bead$^+$ cells (data representative of $n$ = 2 independent differentiations, with two fields of view imaged per well, error bars indicate SEM).

J        Immunocytochemistry of microglia isolated from *Csf-1r*$^{flx/wt}$;*Cx3cr1-Cre*$^−$ and *Csf-1r*$^{flx/wt}$;*Cx3cr1-Cre*$^+$ P0 mouse pup brains. Microglia were stimulated with LPS and exposed to fluorescent opsonised latex beads for 1 h before fixation and quantification of bead$^+$ cells via microscopy. Cells were stained using MitoTracker(red). Arrowheads (white) indicate bead$^+$ cells.

K        Quantification of mouse microglial phagocytic activity expressed as percentage bead$^+$ cells ($n$ = 4 assays, error bars indicate SEM).

L        Immunocytochemistry of macrophages differentiated *in vitro* from *Csf-1r*$^{flx/wt}$;*Cx3cr1-Cre*$^−$ and *Csf-1r*$^{flx/wt}$;*Cx3cr1-Cre*$^+$ mouse bone marrow. Macrophages were stimulated with LPS and exposed to fluorescent opsonised latex beads for 1 h before fixation and quantification of bead + cells via microscopy. Cells were stained using MitoTracker (red) and DAPI (blue). Arrowheads (white) indicate bead$^+$ cells.

M        Quantification of mouse macrophage phagocytic activity expressed as percentage bead$^+$ cells. (*$P$ = 0.024, Student's $t$-test with Welch's correction, $n$ = 4 assays, error bars indicate SEM).

treatment (Fig 5B). To control for the effects of PLX3397 itself on endothelial cells, it was also added to endothelial cells. Although ZO1 and occludin displayed reduced protein levels at 24 h in response to CSF-1R inhibition alone, the critical BBB-associated tight junction protein claudin-5 displayed a microglia-specific response to treatment, with Lo-BVCM reducing protein levels at 24 h (Fig 5C and D). Hi-BVCM induced a downregulation of *Tjp1* expression, as did PLX3397 itself (Fig 5D). Contrastingly, *Ocln* appeared to have been upregulated by untreated microglia conditioned media, an effect which was reduced in response to Lo-BVCM, and ablated completely in the case of Hi-BVCM.

To investigate the interactions in the context of both microglia and endothelial cells having *Csf-1r* heterozygosity, we isolated microglial cells from normal mice (WT) or those lacking a single *Csf-1r* allele in their microglial cells. We collected conditioned medium from these cells in addition to collecting conditioned medium from WT microglial cells where CSF-1R was inhibited using PLX3397. Subsequently, this microglial conditioned medium (MCM) was added to confluent monolayers of primary brain microvascular endothelial cells isolated from WT mice, or mice lacking a single *Csf-1r* allele in their endothelial cells (Fig 5E). Intriguingly, MCM isolated from microglial cells lacking one *Csf-1r* allele could induce a potent downregulation of claudin-5 protein levels (Fig 5F and G). Endothelial cells lacking a single *Csf-1r* allele did not manifest a decreased level of claudin-5 in response to MCM treatment; however, levels of other tight junction components such as occludin and ZO-1 were decreased in a CSF-1R-independent manner (Fig 5C).

To elaborate on the genotype-specific responses observed, we examined transcriptional changes in WT and *Csf-1r*$^{+/−}$ endothelial cells treated with MCM produced by either WT or *Csf-1r*$^{+/−}$ microglia. Expression of TJ components claudin-5, occludin and tricellulin were found to be significantly reduced *Csf-1r*$^{+/−}$ endothelial cells

following treatment with *Csf-1r*$^{+/−}$ MCM (Fig 5H–O). Wild-type endothelial cells also displayed a resistance to *Csf-1r*$^{+/−}$ MCM, maintaining tight junction gene expression (Fig 5H–O). With regard to expression of CSF-1R pathway components, transcription of both *Csf-1* and *Il34* was elevated in *Csf-1r*$^{+/−}$ endothelial cells and in the case of *Il34*, *Csf-1r*$^{+/−}$ MCM induced an upregulatory response in *Csf-1r*$^{+/−}$ endothelial cells. In contrast, *Csf-1r*$^{+/+}$ MCM inhibited *Il34* expression in *Csf-1r*$^{+/−}$ endothelial cells, suggesting microglia may regulate endothelial *Il34* expression in a CSF-1R dependent manner (Fig EV4A and C). The response of the macrophage inhibitory factor (*Mif*) expression was similar to that of *Il34*, with an upregulation of endothelial *Mif* expression only observed in *Csf-1r*$^{+/−}$ endothelial cells treated with *Csf-1r*$^{+/−}$ MCM (Fig EV4F). Upregulation of *Mif* expression has been implicated in the activation of microglial-mediated inflammatory processes and has recently been proposed to be a biomarker of Alzheimer's disease (Zhang *et al*, 2016, 2019). Furthermore, MIF expression has been demonstrated to enhance tight junction breakdown and blood–brain barrier permeability in stroke (Liu *et al*, 2018), and MIF-mediated processes are reported to be significantly altered in ALSP frontal cortex white matter (Kempthorne *et al*, 2020). The restriction of these transcriptional changes in TJ components, *Il34* and *Mif* to *Csf-1r*$^{+/−}$ endothelial cells responding to soluble factors from *Csf-1r*$^{+/−}$ microglia indicates CSF-1R as a key regulator of microglial-endothelial crosstalk.

## CSF-1R mutations and inhibition induce blood–brain barrier disruption

Given the evidence of CAA and the cerebrovascular pathology apparent in each post-mortem ALSP case we examined, we stained post-mortem ALSP brain sections for claudin-5, a key mediator of tight junction function at the BBB. In areas of dense amyloid-beta

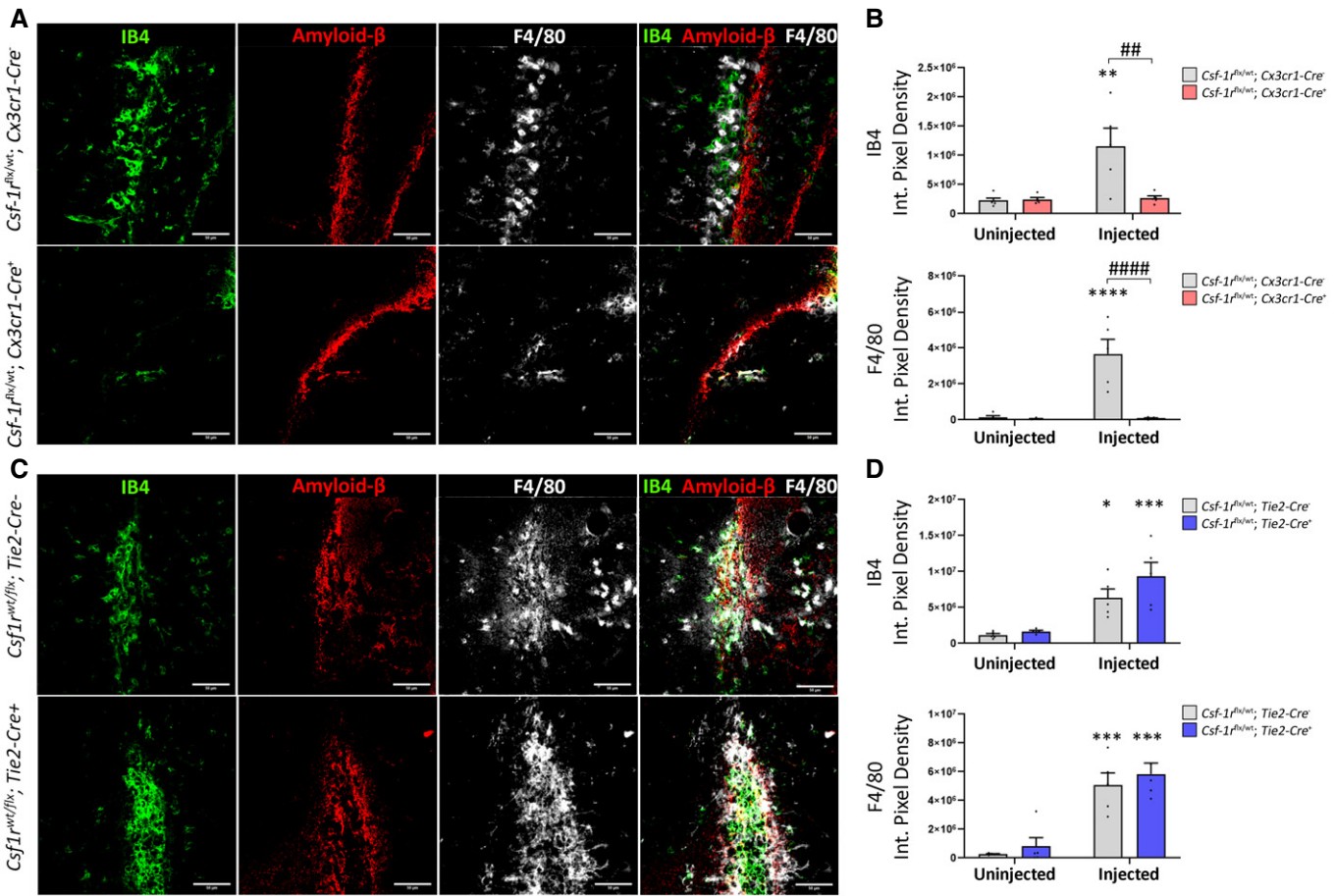

**Figure 4. CSF-1R expression in endothelial cells and macrophages affects response to amyloid injection.**

A Immunohistochemistry of $Csf\text{-}1r^{flx/wt};Cx3cr1\text{-}Cre^{-}$ (top) and $Csf\text{-}1r^{flx/wt};Cx3cr1\text{-}Cre^{+}$ (bottom) mice unilaterally injected with Aβ1-42 in the hippocampus and stained for IB4 (green), F4/80 (white) and Aβ (red). Scale bars indicate 50 μm.

B Quantification of F4/80 and IB4 immunopositivity following intrahippocampal injection of Aβ1-42. Quantification of immunopositivity in both injected and uninjected hippocampi of $Csf\text{-}1r^{flx/wt};Cx3cr1\text{-}Cre^{-}$ and $Csf\text{-}1r^{flx/wt};Cx3cr1\text{-}Cre^{+}$ mice. (Two-way ANOVA with Sidak's test for multiple comparisons. Asterisks (*) indicate comparison to immunopositivity values of the uninjected hippocampus, obliques (#) indicate comparison between $Csf\text{-}1r^{flx/wt};Cx3cr1\text{-}Cre^{-}$ and $Csf\text{-}1r^{flx/wt};Cx3cr1\text{-}Cre^{+}$ mice, ## or **$P < 0.003$, #### or ****$P < 0.0001$, $n = 5$ mice per group, error bars indicate SEM.)

C Immunohistochemistry of $Csf\text{-}1r^{flx/wt};Tie2\text{-}Cre^{-}$ (top) and $Csf\text{-}1r^{flx/wt};Tie2\text{-}Cre^{+}$ mice (bottom) mice unilaterally injected with Aβ1-42 in the hippocampus and stained for IB4 (green), F4/80 (white) and Aβ (red).

D Quantification of F4/80 and IB4 immunopositivity following intrahippocampal injection of Aβ1-42. Quantification of immunopositivity in both injected and uninjected hippocampi of $Csf\text{-}1r^{flx/wt};Tie2\text{-}Cre^{-}$ and $Csf\text{-}1r^{flx/wt};Tie2\text{-}Cre^{+}$ mice. ($n = 5$ mice per group. Two-way ANOVA with Sidak's test for multiple comparisons. Asterisks (*) indicate comparison to immunopositivity values of the uninjected hippocampus, *$P < 0.01$, ***$P < 0.0005$, error bars indicate SEM).

deposition around blood vessels, we observed a non-linear distribution of claudin-5 (Fig 6A) in tandem with extravasation of IgG (Fig 6 B) and fibrinogen (Fig 6G) suggestive of BBB disruption. Perivascular localisation of CD68- and CD163-positive cells (Fig 6E and F) indicated a recruitment of macrophages to the vasculature. Added to this, there were also large swathes of GFAP immunoreactivity (Fig 6C and D) in areas of CAA, suggestive of perivascular astrogliosis absent in age-matched controls (Appendix Fig S2). Given the very clear cerebrovascular pathology observed in these post-mortem cases, we conducted dynamic contrast-enhanced MRI (DCE-MRI) in a genetically diagnosed patient with ALSP who had recently become symptomatic (ΔA781_N783). While the patient had clear periventricular FLAIR enhancements, there was also clear evidence of BBB disruption in the cortical and deep white matter regions of the brain

(Fig 6H and I). Susceptibility weighted imaging (SWI) showed evidence of iron deposition in the putamen and in a similar pattern to BBB disruption (Fig 6J).

Given the dysregulated BBB phenotype in life and in postmortem ALSP samples and as CSF-1R can differentially regulate endothelial–microglial crosstalk, we wanted to elucidate the impact of disrupted CSF-1R signalling in brain endothelial cells. In this regard, a dose-dependent decrease in claudin-5 and occludin levels were observed at 24-h post-stimulation of cells with a known CSF-1R inhibitor PLX3397 (Fig EV5A and B). This decrease in tight junction components was accompanied by a concomitant decrease in transcription of both $Cld5$ and $Ocln$ at 24 h, which is maintained through 48 h in the case of Ocln. PLX3397 treatment resulted in an increase in paracellular flux of FD-4 across a polarised monolayer of

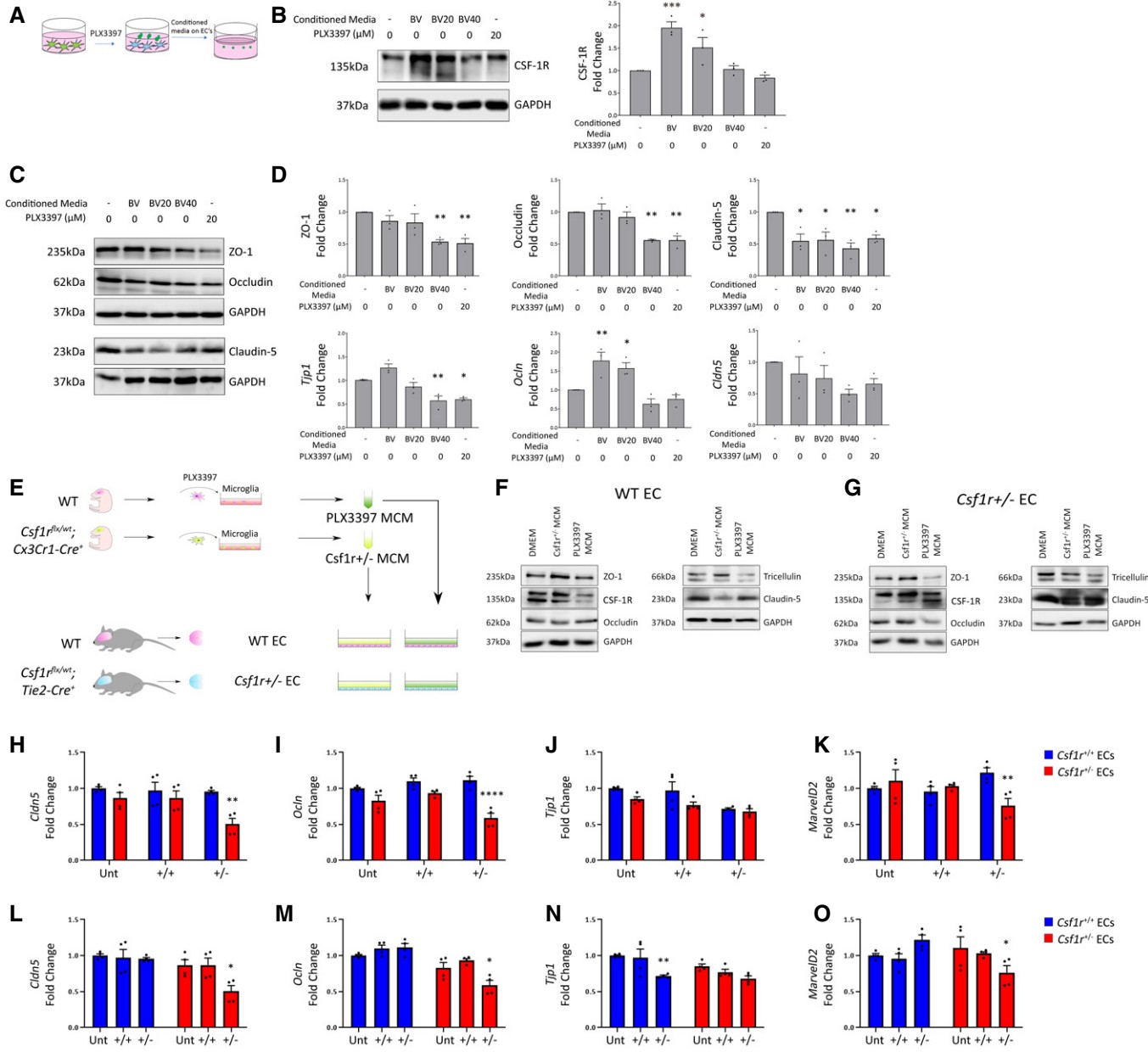

**Figure 5. Endothelial–macrophage crosstalk regulates CSF-1R expression and TJs.**

A  Schematic for experimental design.

B  Western blot for CSF-1R in b.End3 cells treated with different BV2 conditioned media or PLX3397, with corresponding densitometry (right). ($n = 3$ independent BV2 media conditionings and b.End3 treatments, one-way ANOVA with Dunnett's post-test for multiple comparisons, $*P < 0.05$, $***P < 0.0005$, error bars indicate SEM).

C  Western blot for tight junction proteins in b.End3 cells treated with unconditioned, microglia conditioned media or PLX3397.

D  Densitometry of tight junction protein changes (top) and transcriptional changes in *Tjp1*, *Ocln* and *Cld5* (bottom) following conditioned media treatments or 20 μM PLX3397. ($n = 3$ independent BV2 media conditionings and b.End3 treatments, one-way ANOVA with Dunnett's post-test for multiple comparisons, $*P < 0.05$, $**P < 0.009$, error bars indicate SEM).

E  Schematic for experimental design.

F  Western blot of WT endothelial cells treated with unconditioned or microglia conditioned media (MCM) produced from *Csf-1r⁺/⁻* or PLX3397-inhibited microglia.

G  Western blot of *Csf-1r⁺/⁻* endothelial cells treated with unconditioned, *Csf-1r⁺/⁻* MCM or PLX3397 MCM.

H–K  qPCR of wild type (blue) or *Csf-1r⁺/⁻* (red) endothelial cells treated with control, *Csf-1r⁺/⁺* MCM or *Csf-1r⁺/⁻* MCM. Statistical analyses of inter-genotype changes. ($****P < 0.0001$, $**P < 0.005$, Scatter plots represent technical replicates of $n = 2$ independent primary cell isolations and microglia conditionings, Two-way ANOVA with multiple comparisons and Sidak's post-test, error bars indicate SEM).

L–O  qPCR of wild type (blue) or *Csf-1r⁺/⁻* (red) endothelial cells treated with control, *Csf-1r⁺/⁺* MCM or *Csf-1r⁺/⁻* MCM. Statistical analyses of changes relative to respective untreated Control. ($**P < 0.005$, $*P < 0.05$. Scatter plots represent technical replicates of $n = 2$ independent primary cell isolations and microglia conditionings, two-way ANOVA with multiple comparisons and Sidak's post-test, error bars indicate SEM).

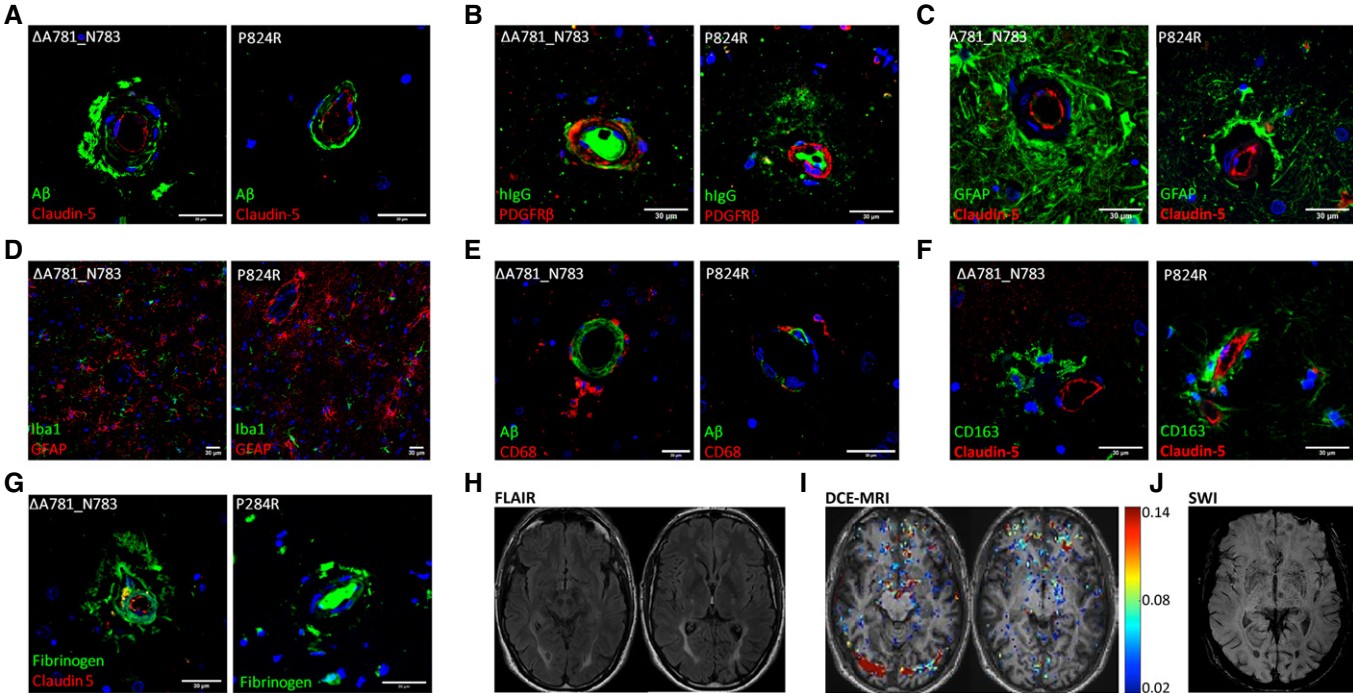

**Figure 6.  Loss of CSF-1R reduces BBB integrity.**

A–C    (A) IHC of ALSP patient post-mortem tissue for amyloid-β (green) and CLD5 (red), (B) human IgG (green) and PDGFRβ (red), (C) GFAP (green) and CLD5 (red). ΔA781_N783 and P824R indicate the CSF-1R variant present.

D–H    (D) Iba1 (green) and GFAP (red), (E) amyloid-β (green) and CD68 (red), (F) CD163 (green) and CLD5 (red), (G) fibrinogen (green) and CLD5 (red) and (H) FLAIR imaging of an ALSP patient with the ΔA781_N783 CSF-1R variant. ΔA781_N783 and P824R indicate the CSF-1R variant present.

I    T1-weighted DCE-MRI of an ALSP patient with the ΔA781_N783 CSF-1R variant. Colormetric scale indicates the slope of the quantified Gd-BOPTA accumulation, as an indicator of BBB permeability.

J    Susceptibility weighted imaging (SWI) of an ALSP patient with the ΔA781_N783 CSF-1R variant showing iron deposition.

Data information: Note that Fig 6C and F are shown again in Fig EV1 for ease of comparison.

primary brain endothelial cells (Fig EV5E) and accompanying down-regulation of *Cld5* (Fig EV5D). While endothelial expression of CSF-1R is disputed, these data provide a tentative link between CSF-1R inhibition and tight junction integrity. Further *in vivo* characterisation of the BBB in the endothelial *Csf1r*$^{-/-}$ mouse will be needed to directly attribute endothelial CSF-1R loss with a corresponding loss in TJ integrity within the cerebrovasculature.

## Discussion

Within the healthy brain, microglia are the primary cells that respond to CSF-1 and IL-34. Following ligation to CSF-1R, the ligand is taken into the cell, reducing the extracellular concentration and preventing long-term local CSF-1R activity. Within the brain of an individual with ALSP however, microglia will constitutively express 50% less functional CSF-1R. This likely results in a CNS saturated with IL-34 and CSF-1 and increased levels of CSF-1 have previously been observed in the CSF-1R-knockout mouse and indeed in some mouse models of AD (Dai *et al*, 2002; Laske *et al*, 2010). This increased pool of CSF-1R ligands may also provide a prolonged opportunity for other cells expressing the receptor to bind ligand and signal through CSF-1R. Therefore, to date, a major focus has

been to understand and potentially target CSF-1R signalling in microglial cells in the context of neuropathologies such as AD and epilepsy.

Here, we sought to understand the role of CSF-1R signalling in microglial cells in the context of haploinsufficiency of CSF-1R. We were, however, surprised to find that microglial cells seemed to adequately tolerate CSF-1R depletion and it was peripheral macrophages and endothelial cells that appeared to be impacted more profoundly. This suggests that, at least in the context of ALSP, the primary driving force of pathology is at the level of the systemic immune response to CSF-1R depletion. Intriguingly, the development of CAA has been shown to occur in up to 80% of sporadic AD cases; however, the underlying molecular aetiology of Aβ deposition around blood vessels is still far from clear. Here, we show that in genetically diagnosed cases of ALSP, CAA development is clear and evident in each case. Added to this, we now confirm that there is no dominant-negative effect of the CSF-1R mutations and they simply lead to a loss of function in kinase activity.

Our findings suggest that inhibiting CSF-1R signalling as previously proposed as a potential therapeutic for AD (Dagher *et al*, 2015; Spangenberg *et al*, 2019) could in fact exacerbate the human pathology. CSF-1R inhibition could potentially impact the phagocytic capacity of macrophages in addition to reducing peripheral

macrophage-mediated clearance of vascular amyloid-β. Secondly, attenuated and aberrant microglial CSF-1R signalling will induce endothelial tight junction re-modelling and BBB permeability as we have shown in post-mortem ALSP cases in addition to contrast-enhanced MRI studies in a genetically diagnosed ALSP patient. Our findings suggest a targeted approach to restoring BBB integrity while also restoring CSF-1R signalling in macrophages could drive a therapeutic response in ALSP patients and other AD-like dementia.

We now propose that symptomatic and genetically diagnosed individuals with ALSP and potentially other forms of AD-like neuropathology will respond to CSF-1R and BBB stabilising approaches to therapy as opposed to CSF-1R inhibition-based strategies as currently proposed. Interestingly, our data also strongly suggest that ALSP patients may be amenable and highly responsive to bone marrow transplantation, with transplantation potentially slowing the course of neurodegeneration. This has recently been evidenced in a small series of case reports where ALSP patients have been shown to stabilise in the months and years after receiving transplantation for alternative perceived reasons (Mochel *et al*, 2019; Gelfand *et al*, 2020). Understanding the molecular aetiology of rare neurodegenerative conditions such as ALSP may be the driving force in identifying novel therapeutic approaches to more common neuropathies. The suggestion of microglial cells being a central driving force of pathology should also be revised to incorporate the critical role played by macrophages and endothelial cells.

# Materials and Methods

### DNA isolation

DNA was isolated from whole blood of live ALSP patients, or from formalin-fixed paraffin embedded (FFPE) tissue. For whole blood, 2 ml of peripheral blood was added to 2 ml of red blood cell (RBC) lysis buffer (0.32 M sucrose, 10 mM Tris, 5 mM $MgCl_2$, 0.75% v/v Triton X-100, pH 7.4). The mixture was inverted and placed on ice for 3 min and spun at 1,500 *g* for 15 m at 21°C. The supernatant was discarded, and pellet resuspended in 2 ml RBC lysis buffer and 6 ml $dH_2O$ before being centrifuged again. The washed pellet was resuspended in proteinase K digest buffer (20 mM Tris, 4 mM $Na_2EDTA$, 100 mM NaCl, 1% w/v SDS, 0.2 mg/ml proteinase K pH 7.4) and incubated at 55°C overnight. 4 ml of 5.3 M NaCl was added to the completed digest followed by 0.2 volumes of chloroform. Tubes were shaken, centrifuged, and the upper aqueous layer moved to a new tube. DNA was isolated using alcohol precipitation followed by resuspension in nuclease free water.

FFPE brain tissue was incubated with gentle shaking in xylene for 15 min at 21°C, centrifuged at 1,500 *g* for 5 m at 21°C, and supernatant was removed and replaced with fresh xylene. This was repeated twice before pelleted tissue was rehydrated by incubating in decreasing concentrations of ethanol, 100, 70, 50% before incubating in $dH_2O$. As above, the pelleted tissue was incubated overnight in proteinase K digest buffer and DNA was isolated.

100 ng of DNA was amplified by PCR in a volume of 25 μl using 1 × reaction buffer, 200 μM each of dNTPs, 0.2 μM of forward and reverse primers, and 1.25 units of DNA Taq polymerase under the following conditions: 95°C 5 min; (95°C 1 min; 58°C 1 min; 72°C 1 min) × 34; 72°C 5 min; 4°C hold. This produced an amplified product of 603 bp using the following primers: forward primer 5′-CTCCAGCAGGGACTCCAAAG-3′ and reverse primer 5′-GGATGCCA TAGGACCAGAC-3′. DNA from the above amplification was purified using a QIAquick PCR purification kit (Qiagen) and subjected to direct sequencing using the forward primer (above).

### Generation of CSF-1R-expressing constructs

For *in vitro* work, Native, and ΔA781_N783 CSF-1R cDNA sequences were synthesised using GeneArt (Thermo Fisher Scientific) and cloned into the AgeI/EcoRI site of the pcDNA3-EGFP expression plasmid (Addgene). The P824R expression construct was generated from the native CSF-1R expression plasmid using the Q5® Site-Directed Mutagenesis Kit (New England Biolabs) and the following primers: forward 5′-TGGATGGCCCGAGAGAGCATCTTTG-3′ and reverse 5′-CTTCACAGGCAGGCGGGC-3′.

### MRI

BBB permeability maps were created using the slope of contrast agent concentration in each voxel over time, calculated by a linear fit model as previously described. Thresholds of high permeability were defined by the 95th percentile of all slopes in a previously examined control group. All imaging was performed using a 3T Philips Achieva scanner and included a T1-weighted anatomical scan (3D gradient echo, TE/TR = 3/6.7 ms, acquisition matrix 268 × 266, voxel size: 0.83 × 0.83 × 0.9 mm), T2-weighted imaging (TE/TR = 80/3,000 ms, voxel size: 0.45 × 0.45 × 0.4 mm), FLAIR (TE/TR = 125/11,000 ms, voxel size: 0.45 × 0.45 × 4 mm).

The calculation of pre-contrast longitudinal relaxation time (T10), the variable flip angle (VFA) method was used (3D T1w-FFE, TE/TR = 2.78/5.67 ms, acquisition matrix: 240x184, voxel size: 0.68x0.68x5 mm, flip angles: 2, 10, 16 and 24°). Dynamic contrast-enhanced (DCE) sequence was then acquired (Axial, 3D T1w-FFE, TE/TR = 2.78/5.6 ms, acquisition matrix: 240x184, voxel size: 0.68x0.68x5 mm, flip angle: 6°, Δt = 6.5 s, temporal repetitions: 70, total scan length: 7.6 min). An intravenous bolus injection of the contrast agent gadobenate dimeglumine (Gd-BOPTA, Bracco Diagnostics Inc., Milan, Italy) was administered using an automatic injector after the first three DCE repetitions. All ethical approvals were in place prior to the initiation of studies on human subjects. All experiments conformed to the principles set out in the WMA Declaration of Helsinki and the Department of Health and Human Services Belmont Report.

### Cell line culture

Human embryonic kidney cells (HEK293, ATCC) were cultured in Dulbecco's modified Eagle's medium containing 4,500 mg/l glucose, GlutaMAX and 110.0 mg/ml sodium pyruvate (DMEM) supplemented with 10% foetal bovine serum (FBS) in a 5% $CO_2$ incubator at 37°C. One day before transfection, HEK293 cells were seeded on 12-well plates ($2.5 \times 10^5$ cells per well). The next day, 500 ng of plasmid containing wild-type or variant *CSF-1R* cDNA was transfected per well using Lipofectamine 2000 (Invitrogen).

For signalling experiments, transfected cells were incubated in DMEM without serum for 16 h. IL-34 or CSF-1 was added at 10, 50 or 100 ng/ml and cells were incubated for 10 min. Cells were washed with PBS on ice and proteins were isolated with lysis buffer (50 mM Tris, 150 mM NaCl, 0.5% w/v sodium deoxycholate, 0.1% v/v Triton X-100 and 0.1% w/v SDS) with added cOmplete™, Mini Protease Inhibitor Cocktail (Roche). For treatment with protein turnover regulators, cells were treated 24 h after transfection with rapamycin (0.1, 0.5, 1, 5 and 10 μM), 3-MA (0.1, 0.2, 0.5, 1 mM) or MG132 (0.1, 0.5, 1, 5 μM). Cells were lysed as above at 24 h posttreatment for rapamycin and 3-MA and at 7 h post-treatment for MG132. For the MG132 pre-treated signalling experiment, cells were treated 24 h post-transfection for 1 h in 1 μM MG132 before being treated with 10, 50 or 100 ng/ml CSF-1 for 10 min and lysed as above.

The mouse microglial BV2 cell line was kindly provided by the Lynch lab (TCIN, Trinity College Dublin). Cells were grown in DMEM with 10% FBS in a 5% $CO_2$ incubator at 37°C, and plated in 1 ml of media at $1.25 \times 10^6$ cells per T25 flask for media conditioning. Flasks were flooded with 12 ml DMEM, 10% FBS 2 h later. At 24 h postseeding, media was refreshed and BV2 cells grown for 48 h to generate conditioned media. Conditioned media was centrifuged at 300 $g$ for 5 min at 21°C, the supernatant passed through a 0.2 μm sterile filter and added to endothelial cultures.

Mouse brain endothelial cells (Bend.3, ATCC) were cultured in DMEM supplemented with 10% FBS in a 5% $CO_2$ incubator at 37°C. Bend.3 cells were seeded on 12-well plates ($2.5 \times 10^5$ cells per well) and allowed to grow to confluency and treated with BV2-conditioned media for 24 h. Cells were lysed for protein analysis as above, and in TRK Lysis buffer (Omega Bio-tek) for RNA isolation. Transfection with *Csf-1r* Smart Pool (Dharmacon) siRNA was performed 24 h after cell seeding using Viromer Blue transfection reagent (Lipocalyx) and 20 pmol siRNA per well.

## Immunocytochemistry

HEK293 cells were seeded on 1% fibronectin-coated 1.3 mm tissue culture coverslips (Sarstedt) in DMEM, 10% FBS. After plasmid transfection, cells were fixed for 10 min at room temperature with ice-cold methanol, washed twice with PBS and incubated with 5% normal goat serum (NGS) before overnight incubation with polyclonal rabbit anti-CSF-1R (Invitrogen; PA5-25974, 1:100) at 4°C. Cells were then washed twice with PBS and incubated with Cy3-conjugated goat anti-rabbit IgG secondary antibody (1:500; Abcam) for 2 h at room temperature and counterstained with Hoechst 33258 to visualise nuclei.

Macrophages were seeded in poly-L-lysine coated 8-well chamber slides (Ibidi) at $2.8 \times 10^4$ cells/cm$^2$. Macrophages were grown for 72 h, fixed in 4% PFA for 10 m at RT and stained as above with rat anti-human CD68 (Abcam) primary antibody and 594-conjugated goat ant-rat IgG secondary antibody(Invitrogen).

## Immunohistochemistry

FFPE sections of autopsied HDLS brains were de-paraffinised in xylene and rehydrated stepwise through decreasing ethanol concentrations. Sections were permeabilised by incubating in methanol for 15 min at −20°C and blocked in 5% NGS, PBS, 0.1%

Triton X-100 for 45 min. Sections were incubated in primary antibody in PBS, 1% NGS, 0.1% Triton X-100 overnight at 4°C (CD68 (Santa Cruz, sc-20060, 1:100), anti-beta-amyloid AW7 (kindly provided by Dominic Walsh, 1:1,000), CLD5 (Invitrogen, 34-1600, 1:200), CD163 (Novocastra, NCL-CD163, 1:200), GFAP (Sigma, G3893, 1:500), hIgG (Abcam, 97170, 1:300), Fibrinogen (DAKO, F0111, 1:300), Iba1 (Wako, 019-19741, 1:500)). Slides were washed three times in PBS and incubated with 594- and 488-conjugated goat anti-rabbit and anti-mouse secondary antibodies (1:500; Invitrogen) for 3 h at 21°C and counterstained with Hoechst 33258 to visualise nuclei.

For IHC of amyloid-injected brains, mice were killed and the brains quickly removed and embedded in optimal cutting temperature compound (VWR), snap-frozen in liquid nitrogen and stored at −20°C prior to slicing on a cryostat. Mouse brain cryosections (20 μm thick) were post-fixed in ice-cold methanol for 10 min at room temperature and washed three times in PBS. Sections were then incubated with 5% NGS before overnight incubation with primary antibodies at 4°C (F4/80 (Abcam; ab6640, 1:100), anti-beta-amyloid AW7 (kindly provided by Dominic Walsh, 1:1,000). Sections were double-stained with isolectin-IB4-Alexa Fluor 488 1:300 (Life Technologies), to label vessels and macrophages. Following three washes in PBS, sections were incubated with Cy3-conjugated goat anti-rabbit IgG and 405-conjugated goat anti-rat secondary antibody (1:500; Abcam) for 2 h at room temperature, washed three times with PBS and counterstained with Hoechst 33258. Sections were mounted and with Aqua Polymount (Polysciences). Sections were imaged with a Zeiss LSM 710 confocal laser scanning microscope. For quantification, pixel intensity was thresholded uniformly across all images and the ratios of pixel intensity values between the injected and uninjected hippocampus were compared. All image analysis was performed using ImageJ (National Institutes of Health, Rockville, MD, USA).

## SDS–PAGE and Western Blotting

Whole cell lysates were diluted in dH2O and 5X Pierce™ Lane Marker Reducing Sample Buffer (Thermo Fisher) and 10 μg total protein loaded per well for separation by SDS–PAGE. Gels were transferred onto methanol-activated polyvinylidene difluoride (PVDF) membranes (Immobilon-P Transfer Membrane, Merck Millipore) via semi-dry transfer. Transferred membranes were re-activated with methanol and blocked under slight agitation in TBS-Tween-20 containing 3% w/v Marvel non-fat dry milk for 1 h at room temperature. Blocked membranes were washed three times with TBS-Tween-20 and treated with primary antibody overnight at 4°C (CLD5 (Invitrogen, 34-1600, 1:1,000), ZO1 (Invitrogen, 402200, 1:1,000), OCLN (Novus Biotech, NBP1-87402, 1:1,000), Triellulin (Invitrogen, 488400, 1:1,000), GAPDH (Cell Signalling, 2118, 1:4,000), phospho-ERK (Cell Signalling, 9101, 1:2,000), total ERK (Cell Signalling, 9102, 1:1,000), CSF-1R (Abcam, ab221684, 1:1,000), β-ACTIN (Abcam, ab8227, 1:4,000) ). Membranes were washed three times for 5 min in TBS-Tween-20 and incubated with horse radish peroxidase (HRP)-conjugated goat anti-rabbit secondary antibody (Sigma) diluted 1:2,000 in TBS-Tween-20 for 2 h at room temperature. Secondary antibody was removed and membranes were washed four times for 5 min in TBS-Tween-20. Blots were developed by enhanced chemiluminescence (ECL). A 1:1

mix of WesternBright ECL Luminol/enhancer solution and Peroxide Chemiluminescent solution (Advansta) was incubated at room temperature for 2 min before being directly added onto washed blots. The LiCor C-Digit Blot Scanner was used to detect chemiluminescence over a 12 min exposure time.

### FACS

PBMCs isolated from HDLS donor blood were seeded in round bottomed 96-well plates in Roswell Park Memorial Institute (RPMI) media with 10% FBS at $4 \times 10^5$ cells per well and incubated in a 5% $CO_2$ incubator at 37°C overnight. Cells were centrifuged at 300 $g$ for 5 min and blocked for 10 min with 50 μl 10% human AB serum, 1% FBS in PBS. Blocked PBMCs were incubated in Live/Dead Aqua (1:500; Life technologies (Fisher/Invitrogen) L34957 ) for 30 min, washed in PBS and incubated in fluorochrome-labelled primary antibody diluted 1:10 in PBS, 1% FBS for 20 min at 4°C. (Lin FITC (BioLegend 348801), CD3 PerCP (Miltenyi 130-100-458), CD19 APCCy7 (Miltenyi 130-098-073), CD14 PacBlue (Miltenyi 130-098-058). Cells were washed twice with PBS 1% FBS and analysed by flow cytometry immediately. Gating during analysis was based on fluorescence minus one controls. Flow cytometry was carried out on a BD LSRFortessa cell analyser and analysed using FlowJo software (Tree Star).

### Primary mouse cerebral microvessel isolation and culture

Microvessels were isolated from cortical grey matter of experimental mice by collagenase/dispase (Roche) digestion and bovine serum albumin density gradient centrifugation. Purified vessels were seeded onto collagen IV/fibronectin-coated tissue culture plates or Corning (Corning) HTS 24-well Transwell polyester inserts (0.4 μm pore size, vessels from five mouse brains per 3 ml) at high density. Cells were grown in EGM2-MV (Lonza) (with 5 μg/ml puromycin during the first 3 days for endothelial cell selection) for 2–3 weeks until their transendothelial electrical resistance values plateaued.

### Transwell permeability assays

Mouse brain endothelial cells ($5 \times 10^4$ cells per well) were grown to confluence on 1% fibronectin-coated Corning HTS 24-well Transwell polyester inserts with a pore size of 0.4 μm and treated with PLX3397 for 24 h. 200 μl of 1 mg/ml of FITC–4kDa Dextran (Sigma-Aldrich) in EGM2-MV2 was added to the apical chamber of each well, and the cells were incubated at 37°C. Sampling aliquots were taken from the basolateral chamber and replaced with fresh medium every 15 min for 2 h and then transferred to 96-well plates (Nunc). FITC-Dextran fluorescence was determined using a spectrofluorometer (Optima Scientific) at an excitation wavelength of 485 nm and an emission wavelength of 520 nm. Relative fluorescence units were converted to values of nanograms per millilitre, using FITC-Dextran standard curves, and were corrected for background fluorescence and serial dilutions over the course of the experiment. The apparent permeability coefficient (Papp) for each treatment was calculated using the following equation:

$$Papp(cm/s) = (dQ/dT)(A \times C0)$$

where dQ/dT (μg/s) is the rate of appearance of FITC-Dextran on the receiver side after application, A ($cm^2$) is the effective surface area of the insert size, and C0 (μg/ml) is the initial FITC-Dextran concentration on the donor side. dQ/dT is the slope m (y = mx + c) calculated by plotting the cumulative amount (Q) versus time (s).

### Primary mouse microglia isolation and culture

Microglia were isolated as described in Cox et. al (Cox et al, 2015). Whole brains were dissected from *Cx3Cr1-Cre*$^{+/-}$;*Csf-1r*$^{wt/Flx}$ or *Cx3Cr1-Cre*$^{-/-}$;*Csf-1r*$^{wt/Flx}$ littermates, chopped and added to working media (DMEM GlutaMAX containing 10% FBS, and penicillin/streptomycin (10 μg/ml). Brains were homogenised using a 5 ml pipette, passed through a 40 μm sterile mesh filter, and spun at 300 $g$ for 5 min at 21°C. Pelleted cells were resuspended in working media, plated in 1 ml per T25 flask and incubated in a humidified environment at 37°C, 5% $CO_2$. 5 ml of working media was added 3 h later following cell attachment. After 24 h, media was supplemented with 20 ng/ml colony stimulating factor 1 (CSF-1) and 10 ng/ml colony stimulating factor 2 (CSF2) (R&D Systems). Mixed glial cultures were grown for 14 days, changing media every 4 days. On day 14, flasks were wrapped with parafilm and shaken at 110 rpm at 21°C for 1 h to isolate nonadherent microglial cells. Suspended cells were centrifuged at 300 $g$ for 5 min at 21°C seeded at $5 \times 10^4$ cells/$cm^2$ in minimal volumes of working media and grown in a 5% $CO_2$ incubator at 37°C. After 2 h, working media containing any unattached cells was removed and replaced.

For media conditioning, $7.3 \times 10^5$ cells were seeded into 6-well tissue culture plates. Microglia were grown for 24 h and had media replaced and collected 24 h later. Conditioned media was centrifuged, sterile filtered and diluted at a 2:1 ratio in EGM2-MV2 before adding to endothelial cultures.

### BMDM isolation

Femurs and tibiae were removed from *Cx3Cr1-Cre*$^{+/-}$;*Csf-1r*$^{wt/Flx}$ or *Cx3Cr1-Cre*$^{-/-}$;*Csf-1r*$^{wt/Flx}$ adult mice and the fat and muscles cut away to leave clean bones.. A 20-ml syringe was filled with DMEM + Glutamax medium supplemented with 10% FCS and 1% P/S, a 27 G needle attached, and the bone marrow was flushed through the bones into a sterile Petri dish. Aggregates were broken up by passing the cells repeatedly through a 20-ml syringe with a 19 G needle attached. The cells were transferred to a 50-ml tube and centrifuged at 300 $g$ for 5 min. Cells were resuspended in the medium with 25 ng/ml colony stimulating factor CSF-1 and plated into 6 dishes/mouse, with 10 ml of media per mouse. Plates were incubated at 37°C with 5% $CO_2$. Cells were fed on day 3 by adding 10 ml of medium with 25 ng/ml of M-CSF per dish. On day 6, trypsin-EDTA was used to remove the cells from the dishes. Cells were seeded onto 13mm tissue culture coverslips and allowed to grow for 24 h before use.

### PBMC isolation and macrophage differentiation

Whole blood was collected in EDTA-coated tubes and diluted 1:1 with PBS, 2% FBS. 20 ml of the blood/PBS mixture was layered onto 10 ml of Lymphoprep (Stemcell Technologies) and centrifuged at 400 $g$ for 45 min with brake and acceleration set to zero. The

plasma layer was removed and PBMCs collected using a Pasteur pipette and washed twice in PBS, 2% FBS followed by a final wash in RPMI. Cells were resuspended in RPMI, 50% FBS, 10% DMSO and frozen.

To generate macrophages, PBMCs were thawed and transferred to complete RPMI, 20% heat-inactivated FBS, 50 ng/ml CSF-1 (cRPMI) and grown for 48 h, rinsed 3 times with RPMI to remove loosely adhered cells and allowed to grow in cRPMI until cells expanded to confluency, changing media every 2 days. Differentiated macrophages were trypsinised and seeded at $2.8 \times 10^4$ cells/cm$^2$.

### Phagocytosis assay

Microglia, BMDM and PBMC-derived macrophages were treated with 10 ng/ml lipopolysaccharide (LPS) for 24 h prior to being assayed. Latex beads, polystyrene amine modified (yellow-green) (Sigma) were pre-opsonised in FBS (1:5) at 37°C for 1 h. Opsonised beads were diluted 1:10,000 in media and added to cells for 1 h. Cells were washed with ice-cold PBS 3 times to remove surface bound beads and fixed in 4% PFA for 10 min and stained with Mito-Tracker Orange CMTMRos for 30 min. Fixed cells were imaged on a Zeiss LSM 710 confocal microscope. Images were analysed by counting bead$^+$ cells in multiple fields of view over 4 replicates and the data expressed as % bead$^+$ cells per genotype.

### Intrahippocampal AB1-42 injection

Mice (8–12 weeks old) were anaesthetised using a ketamine/metadomidine mixture administered via intraperitoneal injection and placed in a stereotaxic frame. An incision was made to expose the skull, and burr holes were made using a surgical drill either above the dorsal hippocampus or the medial prefrontal cortex (mPFC). AggreSure$^{TM}$ Beta-Amyloid 1–42 (AnaSpec) was reconstituted in 50 mM Tris, 150 mM NaCl, pH 7.2 at 0.25 mg/ml per manufacturer's instructions. A Hamilton syringe was loaded with 10 μl reconstituted amyloid, and the needle was slowly lowered into the dorsal hippocampus: (co-ordinates: A/P = −1.9 mm; M/L = ±1.55 mm; D/V = 1.75 mm). 5.0 μl of Beta-Amyloid 1-42 was then injected at a rate of 0.5 μl per min, and once complete, the needle was left in place for 5 min. Anaesthesia was reversed with an intraperitoneal injection of atipamezole and placed in an incubator until recovered. Mice were sacrificed 3 days post-injection and brains taken for IHC.

### Quantitative real-time PCR

RNA was isolated using the E.Z.N.A. Total RNA Kit I (Omega Biotek) and cDNA synthesis was performed using High Capacity cDNA Reverse Transcription Kit (Applied Biosystems). cDNA was diluted 1:10 with nuclease free water and quantitative RT–PCR was performed on a StepOnePlus (Applied Biosystems) machine using SensiFAST$^{TM}$ SYBR$^®$ Hi-ROX Kit (Bioline) and primers listed in Table 1. Cycle threshold (Ct) values were to β-actin and to untreated controls using the delta cT method.

### Stress and toxicity microarray

HEK293 cells were transfected with native or P824R CSF-1R-expressing plasmids and lysed at 24 h post-transfection in TRK

**Table 1. RT–PCR primer sequences.**

| Gene symbol | Forward primer (5′–3′) | Reverse primer (5′–3′) |
|---|---|---|
| Cld5 | TTTCTTCTATGCGCAGTTGG | GCAGTTTGGTGCCTACTTCA |
| Ocln | ACAGTCCAATGGCCTACTCC | ACTTCAGGCACCAGAGGTGT |
| Tjp1 | GCATGTTCAACGTTATCCAT | GCTAAGAGCACAGCAATGGA |
| Marveld2 | CTGAGAATCTGGGTGTGGT | ACGAGTACGAAGGGGGTCTT |
| Actb | GGGAAATCGTGCGTGACAT | GTGATGACCTGGCCGTCAG |
| Csf-1 | AGTGCTCTAGCCGAGATGTGGT | CAGAGGCCGGGTCACTGCTA |
| Csf-1r | AAGCAGAAGCCGAAGTACCA | GTCCCTGCGCACATATTTCAT |
| Il-34 | CGTACAGCGGAGCCTCATGGAT | CAGCTCGCAGTCCTGCCATTTT |
| Mif | GCCAGAGGGGTTTCTGT | GTTCGTGCCGCTAAAAG |
| Icam1 | TGTCAGCCACCATGCCTTAG | CAGCTTGCACGACCCTTCTA |
| Ptprz1 | AATAGCCCAAAGCAGTCTCC | CCGATCCTTCAGATGACACA |

lysis buffer. RNA was isolated and cDNA synthesised using the RT$^2$ First Strand Synthesis Kit (Qiagen), with DNase I digests and genomic DNA elimination steps included. Native and P824R-transfected cell cDNA were analysed using the RT$^2$ Profiler$^{TM}$ PCR Array Human Stress & Toxicity PathwayFinder (Qiagen) per manufacturer's instructions.

### Statistical analyses

GraphPad Prism 8.0 (GraphPad Software) was used for statistical analyses. Statistical analysis was performed using Student's $t$-test, with significance represented by a $P$ value of $\leq 0.05$. For multiple comparisons, ANOVA was used with a Dunnett post-test and significance represented by a $P$ value of $\leq 0.05$. The omnibus K2 (D'Agostino-Pearson) test was used in GraphPad Prism 8 to test for Gaussian distribution of data. G*Power was used a priori to calculate an appropriate sample size to ensure adequate power for experiments. All animal experiments were conducted in a blinded fashion, with the analysis similarly performed by an experimenter blinded to the conditions. Randomisation was used for any experiments using injected material such as amyloid-β. Littermate controls were used for knockout studies and retroactively genotyped.

### Animals

All studies carried out in the Smurfit Institute of Genetics in Trinity College Dublin (TCD) adhere to the principles laid out by the internal ethics committee at TCD and all relevant national licences were obtained prior to commencement of all studies. All mice were bred on-site in the specific pathogen-free unit at the Smurfit Institute of Genetics in TCD.

## Data and software availability

This study includes no data deposited in external repositories.

**Expanded View** for this article is available online.

**The paper explained**

**Problem**

Leucoencephalopathies are a class of progressive diseases of the white matter of the brain which can be either acquired over life or the product of an inherited genetic mutation. Adult-onset leucoencephalopathy with axonal spheroids and pigmented glia (ALSP) accounts for up to a quarter of reported adult-onset leucodystrophies. Despite the rapid and fatally progressive nature of ALSP, there has yet to be an effective treatment for the disease. Caused by mutations in the *CSF1R* gene, which is essential for microglial viability, ALSP is widely considered to be a microgliopathy. Here, we present an alternative approach to the aetiology of the disease which brings into consideration the blood–brain barrier (BBB) and peripheral macrophage populations.

**Results**

Through examining post-mortem cortical samples, we identified two familial ALSP cohorts heterozygous for novel *CSF1R* mutations and presenting with an additional pathology of cerebral amyloid angiopathy (CAA) and BBB breakdown. With these results indicative of a BBB component to ASLP, we sought to characterise the *CSF1R* mutations driving this novel secondary pathology. Through *in vitro* analyses of variant CSF-1R expression constructs, we demonstrate that these novel mutations in *CSF1R* result in a loss-of-function protein being produced, which is actively targeted for degradation by the UPS. Upon examination of peripheral blood mononuclear cell (PBMC) populations of these ALSP patients, we detected a reduction in differentiated PBMCs. Isolating and producing macrophages from ASLP donors revealed ALSP macrophages to have aberrant morphologies and reduced phagocytic capacity, revealing a deficit in peripheral macrophage function. Using mice heterozygous for *Csf1r*, we show that loss of *Csf1r* in bone-marrow-derived macrophages, but not microglia, negatively impacts phagocytosis. We subsequently show that loss of *Csf1r* in macrophages reduces the macrophage response to brain-derived amyloid-β. In addition to the reduced phagocytosis caused by *Csf1r* loss, this may produce a circulating macrophage population conducive to amyloid-β accumulation over life. Rather than forming senile plaques, this amyloid accumulation was exclusively observed in brain microvessels in ALSP patients. This is likely due to dysregulated BBB tight junction maintenance caused by the crosstalk between endothelial cells and microglia heterozygous for *Csf1r*.

**Impact**

As ALSP is widely considered to be a microgliopathy, therapeutic considerations have been focused on a target within the CNS. There have been recent successes with haematopoietic stem cell transplantation in the treatment of ALSP and other leucodystrophies; however, there is a critical lack of understanding as to how this treatment provides benefit. Here, we propose that restoration of the peripheral immune system facilitates the clearance of aggregates within the brain through replacement of circulating variant-CSF1R expressing macrophages, restoring differentiation, phagocytic capacity, macrophage localisation to regions of amyloid-β deposition. We identify the BBB as a potential hub of pathologic microglial–endothelial signalling which drives the cerebrovascular pathology of the disease and demonstrate in life breakdown of the BBB during early stages of the disease. Intriguingly, targeting systemic macrophages and BBB integrity could also show promise for multiple other neurodegenerative conditions, which currently have no approved therapies.

## Acknowledgements

This work was supported by grants from Science Foundation Ireland (SFI), (12/YI/B2614 and 11/PI/1080), The Irish Research Council (IRC), The Health Research Board of Ireland (HRB), the BrightFocus Foundation. The Campbell lab at TCD is also supported by an SFI Centres grant supported in part by a research grant from SFI under grant number 16/RC/3948 and co-funded under the European Regional Development fund by FutureNeuro industry partners. The Campbell lab is also supported by a European Research Council (ERC) grant, "Retina-Rhythm" (864522). SNS is supported as a senior principal investigator by the Flanders Institute for Biotechnology (VIB, Belgium). We would like to thank the VIB BioImaging Core (Ghent, Belgium) for training, support and access to the instrument park. We thank Charles Murray for animal husbandry.

## Author contributions

CD designed the study, performed experiments and wrote the manuscript. MF conducted neuropathological diagnosis and analysis. CD conducted human MRI studies. KB conducted FACS analysis. EOK conducted human MRI analysis. CG and KB conducted *in vitro* experiments. EK conducted human MRI studies. NB conducted neuropathological diagnosis and analysis. PH isolated bloods from patients. SC diagnosed patients. SS performed crystal structure modelling. SD designed the study and wrote the paper. MC designed the study and wrote the paper. All authors contributed to writing and have approved the final version of the manuscript.

## Conflict of interest

The authors declare that they have no conflict of interest.

## For more information

i   https://www.futureneurocentre.ie/
ii  https://www.tcd.ie/Genetics/research/campbell.php

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
