## [Review Process File · EMBO Molecular Medicine]

Attenuated CSF-1R signalling drives cerebrovascular pathology

Conor Delaney, Michael Farrell, Colin Doherty, Kiva Brennan, Eoin O'Keeffe, Chris Greene, Kieva Byrne, Eoin Kelly, Niamh Bermingham, Paula Hickey, Simon Cronin, Savvas Savvides, Sarah Doyle, and Matthew Campbell

DOI: [10.15252/emmm.202012889](https://doi.org/10.15252/emmm.202012889)

Corresponding authors: Matthew Campbell (matthew.campbell@tcd.ie)

Review Timeline:

Submission Date:	5th Jun 20
Editorial Decision:	4th Aug 20
Revision Received:	27th Oct 20
Editorial Decision:	11th Nov 20
Revision Received:	19th Nov 20
Accepted:	23rd Nov 20

Editor: Zeljko Durdevic

Transaction Report:

4th Aug 2020

Dear Dr. Campbell,

Thank you for the submission of your manuscript to EMBO Molecular Medicine, and please accept my apologies for the delay in getting back to you. We have now received feedback from two of the three reviewers who agreed to evaluate your manuscript. As the referee #1 will unfortunately not be able to return his/her report in a timely manner, we prefer to make a decision now in order to avoid further delay in the process. Should referee #1 provide a report, we will send it to you, with the understanding that we will not ask you extensive experiments in addition to the ones required in the enclosed reports from referee #2 and #3.

As you will see from the reports below, while the referee #3 is overall supportive of the study, referee #2 raises serious criticism that should be addressed in a major revision of the current manuscript. The main focus of the revision should be on 1) providing evidence that endothelial cells express CSF1R by performing direct immunohistochemistry on brain sections or as suggested by the referee #2 you should "de-emphasize the results from CSF1R interventions in endothelial cells" and 2) addressing the difference/similarity between the effects of mutant CSF1R expression versus CSF1R haploinsufficiency in vitro and in regard to the mutation phenotypes observed in CSF1R haploinsufficient mouse model.

Addressing the reviewers' concerns in full will be necessary for further considering the manuscript in our journal. Acceptance of the manuscript will entail a second round of review. Please note that EMBO Molecular Medicine encourages a single round of revision only and therefore, acceptance or rejection of the manuscript will depend on the completeness of your responses included in the next, final version of the manuscript. For this reason, and to save you from any frustrations in the end, I would strongly advise against returning an incomplete revision.

We realize that the current situation is exceptional on the account of the COVID-19/SARS-CoV-2 pandemic. Therefore, please let us know if you need more than three months to revise the manuscript.

I look forward to receiving your revised manuscript.

***** Reviewer's comments *****

Referee #2 (Comments on Novelty/Model System for Author):

In our perspective, the authors make two critical conceptual errors with the use of inadequate model systems and problematic technical quality (including missing quantifications for claimed effects). These issues largely question the strength of the evidence and conclusions drawn.

1st error lies in use of endothelial cells to study the CSF1R receptor signalling. These cells do not express the CSF1 receptor in vivo as supported by numerous public resources of cerebral single cell sequencing.

2nd error is to explain the effects of CSF1R mutations with haploinsufficiency models. Said mutations in heterozygous context lead to degradation of nonmutated allele protein resulting in ablation of signalling. The authors instead of introducing these mutations, use heterozygous knockouts that express naive allele which is not targeted for proteolytic degradation.

For details please refer to the comments for the author's section.

Referee #2 (Remarks for Author):

Summary:

The manuscript by Delaney C. et al. presents previously undescribed cases of novel mutations in the CSF-1R receptor and attempts to explore the direct effect of these mutations on microglial and vascular endothelial cells with in-vitro and in-vivo models. The strongest novelty angle for the manuscript message comes from clinical evidence of cerebrovascular pathology in carriers of novel CSF-1R mutations. The mutation effects observed in Figure 1 are intriguing and the consequences on vascular phenotypes (presented in Fig 6) are indeed very interesting. The authors eloquently attempt to describe the mutation effects on cellular signalling by the introduction of respective constructs to HEK cells. However, further interpretation of the mechanistic part of the manuscript relies on two critical conceptual assumptions which are either: (major point 1) - unsupported by generally available evidence or (major point 2) - inconsistently argued within the manuscript.

Major points:

1). Mechanistic analysis of CSF-1R effects in microglial and endothelial cells is based on the assumption that the CSF1R gene is expressed in both cell types. The assumption that CSF1R is present in endothelial cells in-vivo is critical to interpret results from numerous figures (Fig.4 c,d; Fig. 5 c,d,f,g,h,i,j,k,l,m,n,o; Fig. 6 k,l,m,n). However, publicly available resources of single-cell RNA sequencing support *Csfr1* expression only in microglia but not in endothelial cells.

Allen Atlas:

https://celltypes.brain-map.org/rnaseq/mouse_ctx-hip_smart-seq

Mousebrain.org

<http://mousebrain.org/genesearch.html>

Betsholtzlab.org (cerebrovascular resource)

<http://betsholtzlab.org/VascularSingleCells/database.html>

These resources are the current conceptual reference standard to understand gene expression in brains and would be hard to refute.

The notion that CSF-1R protein is expressed in endothelium comes from authors observations and its detection by Western blotting in immortalised Bend3 cell line (deposited by Montesano R, et al. to ATCC in 1990) (Fig.5 b) and primary mouse endothelial cells (Fig.5 f,g). However, the purification protocol for primary cells used by the authors uses only centrifugation gradients rather than more selective endothelial cell-specific pull-downs or fluorescent reporter sorting. Even these advanced endothelial-specific purifications methods are prone to microglial contamination (see Vanlandewijck, M. et al. Nature 2018).

As a consequence, a large proportion of the manuscript results that refer to CSF1r role in endothelium lacks rational support in-vivo and risks to propagate serious misinterpretations by the readers of the journal.

The vascular phenotypes present in the CSF1R mutation carriers are distinct and clear but more likely due to interaction between vascular and parenchymal cells in the CNS rather than direct effects of Csfr1 within endothelium.

Therefore, I would suggest that the authors: (1) acknowledge in the manuscript the lack of support for Csfr1 expression in endothelial cells in vivo, (2) reconsider the value of their observations and claims based on the Csfr1 presence in endothelium and (3) de-emphasize the results from Csfr1 interventions in endothelial cells e.g. by keeping selected data in the supplementary figures.

2). Interpretation of clinical phenotypes arising from CSF1R mutation with the tools that express one healthy allele relies on the assumption that the effects of haploinsufficiency (e.g. heterozygous Csfr1 knockout mice) are the same as the presence of a mutated allele. However, the data presented by the authors does not support this claim and is inconsistently argued in the manuscript.

On one hand, the authors report that the presence of a mutated receptor leads to degradation of both mutated and native allele proteins. E.g. "co-expression of the native CSF-1R gene being attenuated in the presence of the mutant CSF-1R".

This standpoint is supported by:

- Reduced protein stability for both mutations as argued by authors (Fig. 2 a).
 - Empirical evidence from co-expression of native and mutated receptors in HEK cells at 24 and 48 hours (Fig. 2b)
 - Rescued proteasomal degradation of the mutated receptor by MG132 (Fig. 2j)
 - The dominant (rather than recessive) phenotype of the CSF1R mutations in ALS disease.
- These observations would lead to a conclusion that the presence of a mutated allele gives not only reduced but nullified effect on Csf1r signalling.

On the other hand, the authors use the haploinsufficiency models (Csf1r +/- mice) where the native receptor is still expressed (and should not be targeted for proteasomal degradation) to interpret mutation phenotypes.

This approach is used in Fig. 3 h,j; Fig. 4 a,b,c,d; Fig. 5 e-o; These observations may reflect partially reduced receptor expression rather than mutation effects (lack of signalling).

This inconsistency could be solved by introduction of described human CSFR1 mutations to mouse models, but such efforts would be time consuming. Authors could show comparative CSFR1 levels and its activity in both mutated and heterozygosity context at least within the in-vitro setting to clarify these interpretations.

This issue should be clearly specified and put in context of data interpretation in the manuscript discussion.

Minor points

In addition, the manuscript often presents data in a (1) subjective, (2) non-quantified and (3) non-transparent way.

1). Claims of effects are made without points of reference or controls: e.g. Pathology phenotypes in Fig. 1 c,d,e,f,g,h,i,j. Fig. 6 a,b,c,d,e,f,g,h,i,j. should be shown in reference to tissues without neurodegeneration with similar post-mortem collection interval and age-matched individuals.

These figures have potentially highest informative value within the manuscript. Addressing this issue is critical and would allow clearer interpretation.

2). Claims of effects from in-vitro experiments are made without quantifications or statistical analysis:

e.g. Fig. 2 d,e,f,g.; Fig. 3 c; Fig. 5 b,c,f,g; Fig. 6 k.

Presenting a technical (e.g. reloading of the same lysate e.g. per patient when increasing the n-number is not available)

or, better yet, biological reproduction (repeated in vitro experiment) data would allow for more supported claims.

Quantification of data would be necessary to justify the claims of "increased" or "decreased" effects.

3). Whenever the quantifications are indeed present, the figures often conceal the n number and data distribution by

using column graphs with standard deviation:

e.g. Fig. 3 e,f,g,h,i; Fig. 5 d.

This figure format may be justified when presenting data from a standardised experiment like RT-PCR where the n

numbers are the same throughout the manuscript, yet even there the n numbers are not mentioned in the figure panels

or legends.

Referee #3 (Remarks for Author):

Study by Delaney describes the role of mutations in colony stimulating factor-1 receptor (CSF-1R) in cerebral vascular pathology in the context of amyloid pathology and Alzheimer's disease. The paper is very well structured and beautifully designed. The identification of the families with ALS and characterization of their vascular pathology are the real strengths of this study. The results from cell cultures and animal experimentations are very convincing. The findings are novel and very likely to generate a great interest in the field.

Minor: It would be nice if the results from Figure 2 could be confirmed, at least in part, in a cell type that is relevant to cerebral vasculature or amyloid pathology.

Editor's Comments

The main focus of the revision should be on 1) providing evidence that endothelial cells express CSF1R by performing direct immunohistochemistry on brain sections or as suggested by the referee #2 you should "de-emphasize the results from CSF1R interventions in endothelial cells" and 2) addressing the difference/similarity between the effects of mutant CSF1R expression versus CSF1R haploinsufficiency *in vitro* and in regard to the mutation phenotypes observed in CSF1R haploinsufficient mouse model.

As suggested by the editor and Reviewer 2, we have de-emphasized the results from CSF1R in endothelial cells and we concur that the expression of CSF1R in brain endothelial cells is still yet to be fully resolved. Additionally, we have expanded our analysis of CSF1R expression/haploinsufficiency *in vitro* with quantitative data and believe that our manuscript has been vastly improved. We have included some data in our response related to CSF1R expression in brain endothelial cells simply to highlight that we will continue to explore whether or not CSF1R is actually expressed in these cells.

Reviewer's comments

Referee #2 (Comments on Novelty/Model System for Author):

In our perspective, the authors make two critical conceptual errors with the use of inadequate model systems and problematic technical quality (including missing quantifications for claimed effects). These issues largely question the strength of the evidence and conclusions drawn. 1st error lies in use of endothelial cells to study the CSF1R receptor signalling. These cells do not express the CSF1 receptor *in vivo* as supported by numerous public resources of cerebral single cell sequencing.

2nd error is to explain the effects of CSF1R mutations with haploinsufficiency models. Said mutations in heterozygous context lead to degradation of nonmutated allele protein resulting in ablation of signalling. The authors instead of introducing these mutations, use heterozygous knockouts that express naive allele which is not targeted for proteolytic degradation.

For details please refer to the comments for the author's section.

Referee #2 (Remarks for Author):

Summary:

The manuscript by Delaney C. et al. presents previously undescribed cases of novel mutations in the CSF-1R receptor and attempts to explore the direct effect of these mutations on microglial and vascular endothelial cells with *in-vitro* and *in-vivo* models. The strongest novelty angle for the manuscript message comes from clinical evidence of cerebrovascular pathology in carriers of novel CSF-1R mutations. The mutation effects observed in Figure 1 are intriguing and the consequences on vascular phenotypes (presented in Fig 6) are indeed very interesting. The authors eloquently attempt to describe the mutation effects on cellular signalling by the introduction of respective constructs to HEK cells. However, further interpretation of the mechanistic part of the manuscript relies on two critical conceptual assumptions which are either: (major point 1) - unsupported by generally available evidence or (major point 2) - inconsistently argued within the manuscript.

Major points:

1). Mechanistic analysis of CSF-1R effects in microglial and endothelial cells is based on the assumption that the CSF1R gene is expressed in both cell types. The assumption that CSF1R is present in endothelial cells *in-vivo* is critical to interpret results from numerous figures (Fig.4 c,d; Fig. 5 c,d,f,g,h,i,j,k,l,m,n,o; Fig. 6 k,l,m,n). However, publicly available resources of single-cell RNA sequencing support *Csfr1* expression only in microglia but not in endothelial cells.

Allen Atlas: https://celltypes.brain-map.org/rnaseq/mouse_ctx-hip_smart-seq

Mousebrain.org <http://mousebrain.org/genesearch.html>

Betsholtzlab.org (cerebrovascular resource)

<http://betsholtzlab.org/VascularSingleCells/database.html>

These resources are the current conceptual reference standard to understand gene expression in brains and would be hard to refute.

The notion that CSF-1R protein is expressed in endothelium comes from authors observations and its detection by Western blotting in immortalised Bend3 cell line (deposited by Montesano R, et al. to ATCC in 1990) (Fig.5 b) and primary mouse endothelial cells (Fig.5 f,g). However, the purification protocol for primary cells used by the authors uses only centrifugation gradients rather than more selective endothelial cell-specific pull-downs or fluorescent reporter sorting. Even these advanced endothelial-specific purifications methods are prone to microglial contamination (see Vanlandewijck, M. et al. Nature 2018). As a consequence, a large proportion of the manuscript results that refer to *Csfr1* role in endothelium lacks rational support *in-vivo* and risks to propagate serious misinterpretations by the readers of the journal.

The vascular phenotypes present in the CSF1R mutation carriers are distinct and clear but more likely due to interaction between vascular and parenchymal cells in the CNS rather than direct effects of *Csfr1* within endothelium.

Therefore, I would suggest that the authors: (1) acknowledge in the manuscript the lack of support for *Csfr1* expression in endothelial cells *in vivo*, (2) reconsider the value of their observations and claims based on the *Csfr1* presence in endothelium and (3) de-emphasize the results from *Csfr1* interventions in endothelial cells e.g. by keeping selected data in the supplementary figures.

We thank Reviewer #2 for these comments and we fully acknowledge the need for further validation of CSF-1R expression within the mouse brain endothelium *in vivo*. In that regard and as suggested, we have moved data from Figure 6, PLX3397 treatment of endothelial cells and the effects of direct CSF-1R inhibition on endothelial tight junction expression, to Extended View Figure 7 to de-emphasise the direct effects of CSF-1R on cerebrovascular endothelial TJ maintenance. We have also alluded to this in the introduction.

While not part of the revised manuscript, we will continue to explore the expression pattern of CSF-1R in BBB associated endothelial cells. As alluded to by Reviewer#2, it has been previously reported in primary endothelial cells and mouse CNS endothelium (¹), in addition to murine neurons (²), however we acknowledge the problems associated with the various protocols as outlined.

Interestingly, human endothelial cells appear to express CSF-1R by direct immunohistochemistry and RNA-seq in the Human Protein atlas (<https://www.proteinatlas.org/ENSG00000182578-CSF1R/tissue/cerebral+cortex#img>).

Added to this, we have directly enquired through personal communication with the Betsholtz lab regarding CSF-1R expression in their datasets. We have included below some scRNAseq data indicating a subset of Iba1⁺ endothelial cells (Figure 1) from microvascular samples that do express *Csf1r* in the mouse brain (Figure 2). Clearly however, microglia express huge amounts of CSF-1R when compared to EC's.

Gene symbol: Aif1

Figure A: detailed expression in each cell (scRNAseq)

Figure 1: scRNA reads demonstrating low AIF1 (*Iba1*) expression in the Brain Endothelial Cell (BEC) populations of the scRNA dataset

Gene symbol: Csf1r

Figure A: detailed expression in each cell (scRNAseq)

Figure 2: scRNA reads demonstrating *Csf1r* expression in the Brain Endothelial Cell (BEC) populations of the scRNA dataset

Furthermore, we have also isolated mouse microvessels and performed direct IHC for CSF-1R (Figure 3). While we do show a degree of signal overlap between claudin-5 positive EC's and CSF1R, we are showing this data simply to suggest that future work will be needed to explore whether CSF-1R is definitively expressed or not in the cerebrovascular endothelium.

Figure 3. IHC for CSF-1R (red), Claudin-5 (green) and CD31 (blue) in isolated mouse cerebral microvessels. Isotype control indicates microvessels which underwent identical staining processes, in the absence of primary antibody. $n = 3$ indicates vessels isolated from 3 separate mice.

2). Interpretation of clinical phenotypes arising from CSF1R mutation with the tools that express one healthy allele relies on the assumption that the effects of haploinsufficiency (e.g. heterozygous *Csf1r* knockout mice) are the same as the presence of a mutated allele. However, the data presented by the authors does not support this claim and is inconsistently argued in the manuscript. On one hand, the authors report that the presence of a mutated receptor leads to degradation of both mutated and native allele proteins. E.g. "co-expression of the native CSF-1R gene being attenuated in the presence of the mutant CSF-1R".

This standpoint is supported by:

- Reduced protein stability for both mutations as argued by authors (Fig. 2 a).
- Empirical evidence from co-expression of native and mutated receptors in HEK cells at 24 and 48 hours (Fig. 2b)
- Rescued proteasomal degradation of the mutated receptor by MG132 (Fig. 2j)
- The dominant (rather than recessive) phenotype of the CSF1R mutations in ALSP disease. These observations would lead to a conclusion that the presence of a mutated allele gives not only reduced but nullified effect on *Csf1r* signalling.

We fully agree with Reviewer #2. We are showing that the mutated allele is one involving loss of function. The data suggests that the variant CSF-1R isoforms found in ALSP are true loss of function, lacking signalling capacity (Figure 2, b - g) while additionally being actively targeted for degradation via the UPS (Figure 2, j). Furthermore, we demonstrate that this increased UPS activity has no impact on cellular stress or toxicity (Supplementary Figure 3). Taken together, this indicates that in the context of ALSP, individuals are functionally

heterozygous for CSF-1R, therefore allowing us to utilise a CSF-1R heterozygous mouse model to further investigate the disease.

We demonstrate that MG132 can rescue variant CSF-1R expression, *without restoring signalling capacity* (supplementary figure 4). MG132 targeted degradation is specific to the variant isoform, indicating that native CSF-1R would not be degraded through this pathway as shown by protein levels being unaffected by MG-132 treatment (Figure 2, j). We similarly directly demonstrate that in the heterozygous state, reduced signalling capacity persists (Figure 2, d, e, f, g) supporting the dominant nature of ALSP as arising from haploinsufficiency, rather than a dominant-negative effect. Again, this supports our use of heterozygous mice.

On the other hand, the authors use the haploinsufficiency models (Csf1r +/- mice) where the native receptor is still expressed (and should not be targeted for proteasomal degradation) to interpret mutation phenotypes. This approach is used in Fig. 3 h,i; Fig. 4 a,b,c,d; Fig. 5 e-o; These observations may reflect partially reduced receptor expression rather than mutation effects (lack of signalling).

We have established the ALSP context to be one of haploinsufficiency. There is no cell stress effect from the variant receptor being targeted for degradation. Additionally, CSF-1R signalling capacity is effectively in the heterozygous state, therefore a heterozygous mouse model is the ideal molecular model of CSF-1R bioactivity in ALSP.

This inconsistency could be solved by introduction of described human CSFR1 mutations to mouse models, but such efforts would be time consuming. Authors could show comparative CSFR1 levels and its activity in both mutated and heterozygosity context at least within the in-vitro setting to clarify these interpretations. This issue should be clearly specified and put in context of data interpretation in the manuscript discussion.

Minor points

In addition, the manuscript often presents data in a (1) subjective, (2) non-quantified and (3) non-transparent way.

1). Claims of effects are made without points of reference or controls: e.g. Pathology phenotypes in Fig. 1 c,d,e,f,g,h,i,j. Fig. 6 a,b,c,d,e,f,g,h,i,j. should be shown in reference to tissues without neurodegeneration with similar post-mortem collection interval and age-matched individuals. These figures have potentially highest informative value within the manuscript. Addressing this issue is critical and would allow clearer interpretation.

The neuropathological features displayed in Figure 1 were detected during autopsy by our colleague and co-author Prof Michael Farrell and the processing of sections used pathognomonic markers for diagnostic purposes. In that regard it is challenging to obtain FFPE non-diseased human cortical tissue, dissected and identically processed for a side-by-side comparison. We fully concur however that a non-diseased comparison is important and we have now provided tight junction, CD68 and GFAP immunohistochemistry from age-matched non-diseased, non-demented flash-frozen human tissues (See Figure EV6). The

absence of cerebrovascular and neurodegenerative pathologies in the healthy brain is well established in the literature. The presence of the hallmark pathology of ALSP within the white matter itself is criteria for diagnosis, being a qualitative assessment rather than quantitative. Testing for the pathognomic markers of axonal spheroids, demyelination and amyloid is not routinely performed in tissue obtained during non-demented or cognitively healthy human autopsy.

2). Claims of effects from in-vitro experiments are made without quantifications or statistical analysis: e.g. Fig. 2 d,e,f,g; Fig. 3 c; Fig. 5 b,c,f,g; Fig. 6 k. Presenting a technical (e.g. reloading of the same lysate e.g. per patient when increasing the n-number is not available) or, better yet, biological reproduction (repeated in vitro experiment) data would allow for more supported claims. Quantification of data would be necessary to justify the claims of "increased" or "decreased" effects.

3). Whenever the quantifications are indeed present, the figures often conceal the n number and data distribution by using column graphs with standard deviation: e.g. Fig. 3 e,f,g,h,i; Fig. 5 d.

This figure format may be justified when presenting data from a standardised experiment like RT-PCR where the n numbers are the same throughout the manuscript, yet even there the n numbers are not mentioned in the figure panels or legends.

We acknowledge this oversight on our part and we now included fully quantified figures to represent our replicates, representation of data and statistical analyses in full. This is now reflected throughout all figures.

Referee #3 (Remarks for Author):

Minor: It would be nice if the results from Figure 2 could be confirmed, at least in part, in a cell type that is relevant to cerebral vasculature or amyloid pathology.

We thank reviewer #3 for their positive feedback in support of this study. HEK293 cells were used for molecular characterisation of the native and variant CSF-1R isoforms as they lack endogenous CSF-1R expression. The conclusions we have drawn from the experiments detailed in Figure 2 are indeed limited in terms of relevance to amyloid pathology. However, HEK293 cells were used solely to model the biochemical activities of the native and variant CSF-1R isoforms. Experiments with relevance to amyloid pathology (phagocytosis, endothelial function) were indeed performed in relevant cell types, namely endothelial and macrophage/microglial cells.

References

1. Jin S, Sonobe Y, Kawanokuchi J, et al. Interleukin-34 restores blood-brain barrier integrity by upregulating tight junction proteins in endothelial cells. *PLoS ONE*. 2014;9(12):1-11. doi:10.1371/journal.pone.0115981
2. Luo J, Elwood F, Britschgi M, et al. Colony-stimulating factor 1 receptor (CSF1R) signaling in injured neurons facilitates protection and survival. *Journal of Experimental Medicine*. 2013;210(1):157-172. doi:10.1084/jem.20120412

11th Nov 2020

Dear Dr. Campbell,

Thank you for the submission of your revised manuscript to EMBO Molecular Medicine. I am pleased to inform you that we will be able to accept your manuscript pending the following final amendments:

Please implement all adjustments suggested by the referee #2.

***** Reviewer's comments *****

Referee #2 (Comments on Novelty/Model System for Author):

Highest novelty value within the manuscript lies in the discovery of new CSFR1 mutations and mechanistic descriptions of how they affect the downstream signalling of this receptor.

The technical quality of the pathological assessment is graded at medium. The authors do provide histological and mechanistic assessments yet do not provide tissue controls as a reference to the observed disease-associated pathology.

The medical impact is graded at medium. The authors do describe pathological and mechanistic consequences of the new mutations yet this is largely an observatory study and they do not propose a conceptual or experimental framework for treatment strategies.

The employed model systems are more adequate in the revised manuscript since the authors contextualise the differences in CSFR1 expression between microglia and endothelial cells. Moreover the authors convincingly argue for the haploinsufficiency rather than dominant negative phenotype of the mutations.

Referee #2 (Remarks for Author):

In the revised manuscript the authors adequately contextualize the difference of CSFR1 expression between the microglial and endothelial cells which addressed the main issue nr 1.

The authors also convincingly argue for the haploinsufficiency rather than dominant negative phenotype of the mutations which addressed the main issue nr 2.

The minor issue of adequate quantification analysis and data presentation is largely addressed by new densitometry graphs in Fig. 2 and individual datapoints are now visible on graphs. However the pathological assessments are not referenced to non diseased tissues due to unavailable age-matched controls for Fig. 1.

Minor comments for the revised manuscript.

1. In the paragraph "Mutations in CSF-1R attenuate kinase activity and signalling" on page 4 when referencing to Fig 2 I, the authors describe the use of rapamycin as an autophagy inhibitor. I believe they meant 3-methyladenosine.

2. Imaging figures 1 c,d,e,f,g,h,i,j and EV1, do not have scale bars. It would be informative to include those since particularly Fig. 1 presents observations at different magnifications.

The authors performed the requested changes.

Reviewer's Comments - Remarks for Author:

1. In the paragraph "Mutations in CSF-1R attenuate kinase activity and signalling" on page 4 when referencing to Fig 2 I, the authors describe the use of rapamycin as an autophagy inhibitor. I believe they meant 3-methyladenosine.

We thank you for spotting this mistake. This error has been corrected and 3-methyladenosine is now correctly mentioned.

2. Imaging figures 1 c,d,e,f,g,h,i,j and EV1, do not have scale bars. It would be informative to include those since particularly Fig. 1 presents observations at different magnifications.

Scale bars have been added to each of these figures 1 e,f,g,h,i,j and EV1. Figure 1 c and d are macroscopic images for which we were unable to find information to base the scale bars upon.

We are pleased to inform you that your manuscript is accepted for publication.

YOU MUST COMPLETE ALL CELLS WITH A PINK BACKGROUND ↓
PLEASE NOTE THAT THIS CHECKLIST WILL BE PUBLISHED ALONGSIDE YOUR PAPER

Corresponding Author Name: Matthew Campbell

Manuscript Number: EMM-2020-12889